# The role of oscillations in grid cells' toroidal topology

**Giovanni di Sarra**[ID][1], **Siddharth Jha**[2], **Yasser Roudi**[ID][1,3]*

**1** Kavli Institute for Systems Neuroscience and Centre for Algorithms in the Cortex, Faculty of Medicine and Health Sciences, Norwegian University of Science and Technology, Trondheim, Norway, **2** W.M. Keck Center for Neurophysics, Department of Physics and Astronomy, University of California Los Angeles, Los Angeles, California, United States of America, **3** Department of Mathematics, King's College London, London, United Kingdom

* yasser.roudi@kcl.ac.uk

**Data availability statement:** The codes used for the analysis reported in this paper can be accessed from https://github.com/gdisarra/Oscillations_toroidal_topology. The data analysed here as well as some of the codes are

## Abstract

Persistent homology applied to the activity of grid cells in the Medial Entorhinal Cortex suggests that this activity lies on a toroidal manifold. By analyzing real data and a simple model, we show that neural oscillations play a key role in the appearance of this toroidal topology. To quantitatively monitor how changes in spike trains influence the topology of the data, we first define a robust measure for the degree of toroidality of a dataset. Using this measure, we find that small perturbations (~100 ms) of spike times have little influence on both the toroidality and the hexagonality of the ratemaps. Jittering spikes by ~100-500 ms, however, destroys the toroidal topology, while still having little impact on grid scores. These critical jittering time scales fall in the range of the periods of oscillations between the theta and eta bands. We thus hypothesized that these oscillatory modulations of neuronal spiking play a key role in the appearance and robustness of toroidal topology and the hexagonal spatial selectivity is not sufficient. We confirmed this hypothesis using a simple model for the activity of grid cells, consisting of an ensemble of independent rate-modulated Poisson processes. When these rates were modulated by oscillations, the network behaved similarly to the real data in exhibiting toroidal topology, even when the position of the fields were perturbed. In the absence of oscillations, this similarity was substantially lower. Furthermore, we find that the experimentally recorded spike trains indeed exhibit temporal modulations at the eta and theta bands, and that the ratio of the power in the eta band to that of the theta band, $A_\eta/A_\theta$, correlates with the critical jittering time at which the toroidal topology disappears.

## Author summary

Gene regulatory networks involve many genes and neural networks include many neurons. The state of these and other biological systems at any given time is thus generally characterized by high dimensional data that are difficult to describe. Surprisingly,

originally from Gardner et al, Nature 2022, and accessed via the links below
https://github.com/erikher/GridCellTorus
https://figshare.com/articles/dataset/Toroidal_topology_of_population_activity_in_grid_cells/16764508

**Funding:** The study was supported by Research Council of Norway Centre for Neural Computation, grant number 223262 (GdS and YR); Research Council of Norway Centre for Algorithms in the Cortex, grant number 332640 (GdS and YR); Research Council of Norway Centre NORBRAIN, grant number 295721 (GdS and YR); The Kavli Foundation (GdS and YR); UCLA graduate fellowship (SJ). The funder had no role in study design, data collection and analysis, decision to publish, or preparation of the manuscript.

**Competing interests:** The authors have declared that no competing interests exist.

however, in some cases, building lower dimensional, yet sufficiently accurate descriptions is possible. Which factors contribute to this possibility and to what degree? Here, we present a quantitative approach to evaluating the role of various aspects of the data in the emergence of low dimensional, topological, features. Topological features are those that are insensitive to certain deformations of an object, e.g. stretching, but not cutting. We apply this approach to the case of toroidal features discovered in the activity of neural populations in the brain. We show that the activity of these neurons exhibits temporal oscillations and that these oscillations play a critical factor in the emergence of the toroidal topology. Since oscillatory modulations are abundant in the dynamics of many biological systems, the approach presented here will pave the way to understanding the emergence of topological features in biological data.

## Introduction

The activity of a large neuronal population at any given time can be described by a high-dimensional vector whose elements represent the activity of each neuron in the population. Using various techniques for dimensionality reduction, recent work shows that in some cases, this population activity resides on some low-dimensional manifold [1–4]. Most recently, Topological Data Analysis (TDA) has emerged as a powerful approach to extract the topological properties of these manifolds [5,6], and has been applied to a variety of systems, from proteins to collaboration networks [7,8]. Given the increasing number of neurons that can be recorded simultaneously from the brain, topological techniques, in particular persistence homology, have also been a remarkable tool for studying high-dimensional neural data [9–15]. For example, using TDA, recent work has shown that the activity of a population of head-directional neurons forms a (topological) circle [10,11] and that the population activity of grid cells in the Medial Entorhinal Cortex (MEC) forms a torus [12].

Grid cells are neurons in mammalian MEC that preferentially fire in localized regions in space called spatial fields [16–18]. When the animal forages in a two-dimensional box, these fields form a hexagonal lattice that tessellates the entire environment. The emergence of this toroidal topology may be explained by continuous attractor networks (CAN) performing path-integration [19–21]. However, such a mechanism ignores several prominent features of neural activity: these networks employ rate-based neurons, thus ignoring the spike generation process and its stochastic nature, do not take into account oscillatory components present in neural activity, and require specifically designed recurrent connectivity; see [17,22]. Both oscillations and stochasticity in spiking play a crucial role in determining correlations between neuronal activity. Since low-dimensional representations arise as a result of these correlations, it is thus likely that oscillations and stochastic spiking both influence the emergence of topological features in grid cell populations. In this paper, we aim at quantifying and understanding this influence.

If population vectors were constructed from ratemaps of neurons, each expressing regularly arranged fields of the same shape and size, the symmetries of these fields will be reflected in the population vectors. Since such symmetries are precisely the symmetries expressed by a topological torus, it is thus expected that when the length of the data and the number of neurons are very large, toroidal topology can be detected from the population vectors. However, imperfections arising from irregularity in the spatial organization of fields [23–25], differences between fields of each single neuron [26–29], and inherent stochasticity of spikes all break such symmetries. In fact, TDA may fail to detect toroidal topology in the presence of

these imperfections [14]. In general, thus, applications of TDA to neural data involve resorting to various pre-processing steps, e.g. smoothing the spike trains, applying Principal Component Analysis (PCA), and performing persistent homology on time points at which the mean population activity is largest [12]. Although these steps are justified for computational purposes, they may have non-trivial effects on the outcome [30–32]. Furthermore, previous work on applying TDA to neural data analysis focuses on making a binary decision, that is, whether a topological structure, e.g., a toroidal or circular topology, is present or not. In reality, however, data often exhibit different *degrees* of similarity to a toroidal topology or other topological structures. Consequently, while previous work demonstrates the feasibility of using TDA for detecting topological structures, it fails to quantify the degree to which different aspects of neural activity contribute to the detected topological features. Developing a biologically plausible model for the appearance of a toroidal topology requires such a quantification.

In TDA, barcodes are often used to identify the number of connected components, circles, and cavities in high-dimensional data [5,6]. Intuitively, the number of long bars in the barcodes related to dimension zero ($H_0$) indicates the number of connected components in the data, in dimension one ($H_1$) the number of circles and in dimension two ($H_2$) the number of cavities. In real life applications, however, the length of the bars is affected by fluctuations and noise in the data making the long versus short separation ambiguous. Furthermore, associating statistical significance to the barcodes is generally considered an open problem [33,34]. In neural data, to determine the number of long bars in a given dimension, the barcodes that emerge from experiments are compared with those from a shuffled version of the data [12,13] and are taken to be significant long bars accordingly. This leads to the binary classification referred to above: the presence of a torus when there is a significant long bar in $H_0$, two in $H_1$ and one in $H_2$, and the absence of a torus otherwise. A more systematic approach that we develop and use here is to associate a continuous valued *degree of toroidality* to the data, ranging from zero to one. This approach can also be used to define similarity of the barcodes to other topological objects, e.g., to define a *degree of sphericity*. Instead of a binary classification, such quantities measure how similar the barcodes of a given dataset are to those of any topological structure.

Using this measure, we first show that if we add temporal jitters to real spike trains, we find a sigmoidal relationship between the degree of toroidality and the size of the jitter. This allows us to define a critical time scale for each module, below which jitters do not have much effect on toroidality, but larger jitters lead to a substantial decay in toroidality. The critical jitter size ranged from ∼100 to ∼500 ms. These are much larger than the single neuron integration time constant (10-20 ms) used in path-integrating CAN models, at which effective connectivity between grid cells exhibits the spatial phase dependence required by these models [35]. At the same time, they are several times smaller than the time it takes for the rat to go from one grid field to another, which is of the order of seconds (3-5 s). This is another relevant time scale, as the transition from one field to the other would presumably correspond to the time it takes for the population activity to return to a given point. On the other hand, the presence of oscillations on the time scales of theta (∼125 ms) and eta (∼250 ms) in the brain is well established [36,37], suggesting that these oscillations may play a key role in the emergence of toroidal topology. We thus applied TDA to a simple model in which spatial fields arise as a combination of an underlying hexagonal lattice, with and without oscillatory modulations of neural activity.

We found that TDA yields quite different barcodes on real data and on Poisson simulations without oscillations. The addition of oscillations greatly increased this similarity, yielding high toroidality in simulations, similar to the real data. Looking at the power spectrum of the

spike trains from the experimental data, we found that spike trains from grid cells did indeed show a dominant theta band (8-10 Hz) modulation. Another peak was also present in the eta band (3-5 Hz), similar to what has recently been reported in the hippocampus [37]. We found that a larger eta-to-theta power ratio in real data was correlated with the appearance of more robust tori, further demonstrating the role that these oscillations play in toroidal topology.

## Results

To estimate the degree of toroidality of the high-dimensional neural activity, our starting point is to define a continuous quantity that ranges from zero (no torus) to one (ideal torus), based on the commonly used Bottleneck distance [5,6]. In the next subsection, we first describe this measure and test it on simulated data from ideal 3 and 6 dimensional toroidal shapes, assessing how it traces deviations from the torus when we add noise to it.

### Defining the degree of toroidality

A standard distance measure between two barcodes $\tau_d$ and $\tau_d'$ is the bottleneck distance $d_B(\tau_d, \tau_d')$; see *Computing the Degree of Toroidality* in Methods and [6]. The bottleneck distance, however, is not directly applicable for measuring toroidality because of reasons that can be understood by inspecting Fig 1. The bottom barcodes in each pair in Fig 1A and 1B only have two long bars, and can thus represent $H_1$ barcodes from a torus. Intuitively, the top barcode in Fig 1B is, however, very far from that of a toroidal topology while that of Fig 1A is much closer. Despite this, the bottleneck distances in both cases are around 0.9. This problem arises because the actual value of the bottleneck distance depends on both the absolute and relative scales of the bars [38,39]. We resolve this problem by taking into account such scale differences. Formally, we define a normalized bottleneck distance as

$$\widehat{d}_B(\tau_d, \tau_d') = d_B\left(\frac{\tau_d}{u(\tau_d)}, \frac{\tau_d'}{u(\tau_d')}\right)$$

where $u(\tau_d)$ is the maximum of the difference in the starting and end points between pairs of bars in $\tau_d$; see Eq. (8) in Methods. The division operation $\tau_d/u(\tau_d)$ means that the starting

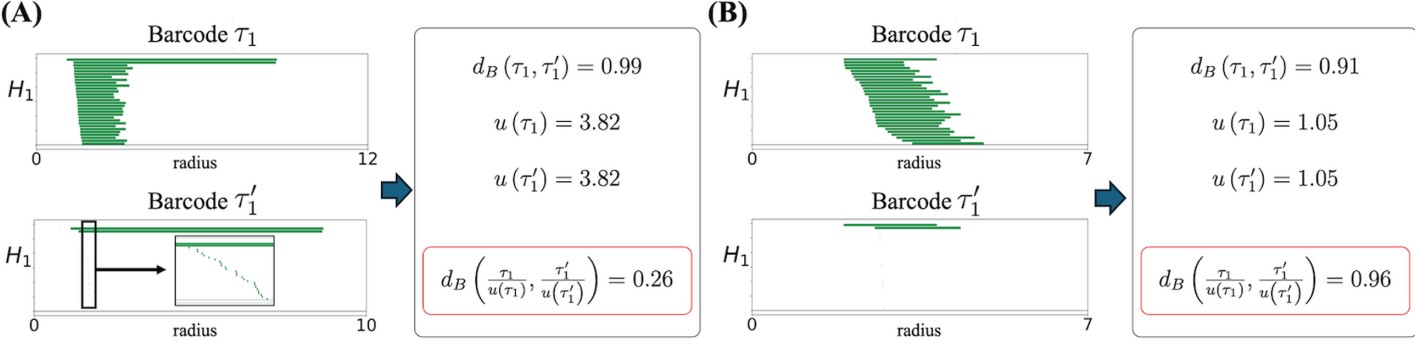

**Fig 1. Computation of normalized bottleneck distance $\widehat{d}_B$.** Procedure for the computation of the normalized bottleneck distance between two barcodes $\tau_1$ and $\tau_1'$ in $H_1$, for **(A)** a barcode consistent with toroidal topology and **(B)** a barcode very different from one expressing toroidal topology. The inset in (A) is a zoomed version showing small and otherwise invisible bars. The value of bottleneck distance is similar in the cases (A) and (B) even though one is much closer to a toroidal barcode than the other. The normalized bottleneck distance, dividing the barcodes by $u(\tau_1)$ and $u(\tau_1')$, takes into account the absolute and relative length of bars, solving this problem.

and end points of all bars in $\tau_d$ are divided by $u(\tau_d)$. This ensures that $0 \leq \widehat{d}_B \leq 1$. As can be seen in Fig 1A and 1B, including this normalization resolves the scale problem.

Using this normalized bottleneck distance, we can now construct a measure of toroidality for a set of barcodes in $H_1$ and $H_2$ as $\Gamma \equiv (\Gamma_1, \Gamma_2)$, where

$$\Gamma_d(\tau_d, \tau_d^{ref}) = 1 - \widehat{d}_B\left(\tau_d, \tau_d^{ref}\right) \tag{1}$$

and $\tau_d^{ref}$ is a reference barcode representing the barcode in dimension $d$ of an idealized torus. We can construct $\tau_1^{ref}$ and $\tau_2^{ref}$ from any set of barcodes (including $\tau_1$ and $\tau_2$ themselves), by retaining the two longest bars in $H_1$ and the single longest bar in $H_2$, respectively; the remaining bars are then kept with the same starting point and their length set to the minimum of all bar lengths in each given dimension. In Fig 1A and 1B the bottom barcodes were in fact constructed in this way from the top barcodes.

To confirm that $\Gamma = (\Gamma_1, \Gamma_2)$ has the properties we expect, we applied it to points on a toroidal parametrization in 3 and 6 dimensions, as we corrupted their coordinates with Gaussian noise (Fig 2).

In 3 dimensions a torus can be parametrically defined by

$$
\begin{aligned}
x &= (c + a \cos v) \cos u \\
y &= (c + a \cos v) \sin u \\
z &= a \sin v
\end{aligned}
\tag{2}
$$

where $v$ and $u$ range in $[0, 2\pi]$, and $a$ and $c$ are the radii of the two circles of the torus. Deviations from this perfect torus were modeled by adding an independent Gaussian noise $\mathcal{N}(0, \delta)$ to each point coordinate in the three-dimensional space. Ideal torus barcodes, $\tau_d^{ref}$, are constructed from those of $\delta = 0$ using the procedure described above in this section. Fig 2 shows that both $\Gamma_1$ and $\Gamma_2$ are close to one for $\delta = 0$; they are smaller than 1 due to the fact that there is a finite number of points on the torus, leading to short living bars.

As the magnitude of noise, $\delta$, increases, both components of $\Gamma$ decrease, exhibiting a sigmoidal dependence on $\delta$. This is shown for two sets of parameters $a$ and $c$ for the 3D tori in Fig 2A and 2B. Similar behavior is observed and shown in Fig 2C when we consider the tori in 6 dimensions described by the parametrization in Methods.

## Degree of toroidality of grid cell populations

Having defined a quantity to measure the similarity of barcodes to those of an ideal torus, we now look at evaluating the degree of toroidality for the grid cells recorded experimentally from [12].

The dataset included data from 3 rats, named R, Q and S. There were 3 modules, R1, R2 and R3, recorded from rat R, two modules, Q1 and Q2, from rat Q and one module from rat S. For the modules recorded from rat R, data from two recording sessions were available. Neurons recorded from each module were divided into pure grid cells and grid cells that, in addition to spatial tuning, also exhibited head-directional tuning. This latter group is referred to as Grid-HD cells. We refer to data from each module in a way that makes explicit the spacing of the different modules, such that, for example $R_{61}^1$ and $R_{58}^1$ refer to the data from module 1 in rat R on two different days (day 1 and day 2), where the spacings were estimated to be 61 cm and 58 cm, respectively. The mean and standard deviations of spacings across neurons from each module and total number of neurons recorded from the module are reported in Table 1.

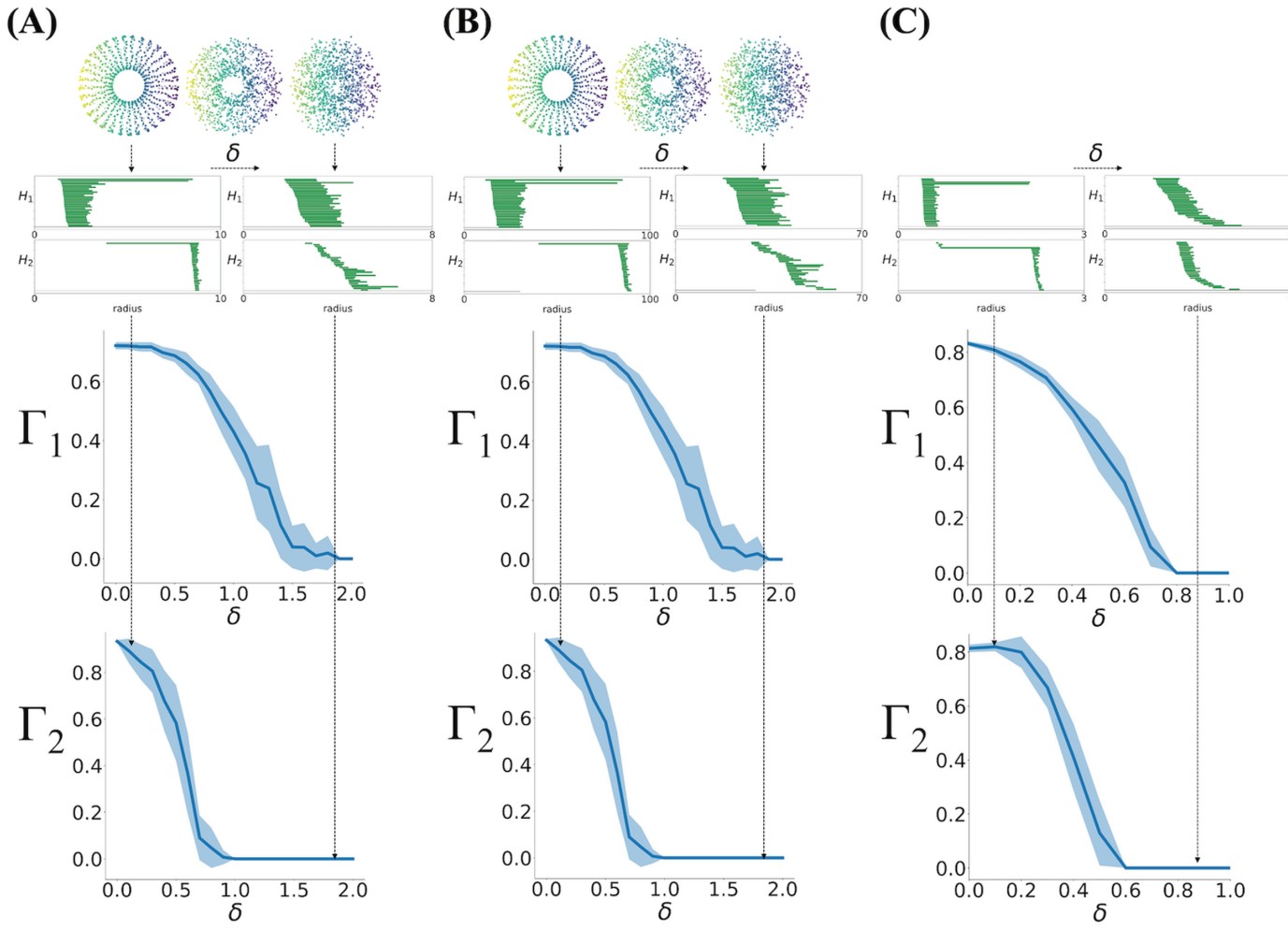

**Fig 2. Sigmoidal relationship between Γ and noise added to simulated tori. (A)** Zero mean Gaussian noise with standard deviation $\delta$ was added to 1200 points on a 3-dimensional torus defined by Eq. (2) with $a = 5$ and $c = 10$, shown on the top row for three increasing values of $\delta$. Persistent homology is performed on this dataset. Barcodes for low (left) and high (right) noise levels are shown in the middle row. The two plots on the bottom show the components of the degree of toroidality, $\Gamma = (\Gamma_1, \Gamma_2)$ versus the size of the noise, $\delta$. The solid line shows the average over 20 realizations of the noise, and the shaded region is the standard deviation. The vertical arrows connects the torus, barcodes and toroidality for two values of $\delta$. **(B)** Same as (A) with $a = 50$ and $c = 100$. **(C)** Same as (A, B) for 20 realizations of data generated on a 6-dimensional torus from Eq (9), in Methods.

Since the data from each module contained both pure grid cells and grid-HD cells (see numbers in Table 1), in Fig 3, we plot $\Gamma_1$ and $\Gamma_2$ both when only pure grid cells were included in the analysis (Fig 3A), and when all recorded cells were used (Fig 3B). In both cases, the reference barcode is constructed directly from the analyzed experimental module, as described in the previous section.

When only pure grid cells are considered, although the barcodes of all six modules pass a statistical significance threshold to be consistent with toroidal topology [12], Fig 3A shows that they clearly exhibit different degrees of toroidality. In particular, $S_{59}$ has $\Gamma_1$ smaller than the other ones, quite far from a torus. Similarly, the degree of toroidality of $R^3_{105}$ is only marginally consistent with a torus. In fact, inspecting the barcodes shows that for both $S_{59}$ (Fig 3C) and $R^3_{105}$ (Fig 3D), more than 2 long bars are likely present in $H_1$. Furthermore, the

**Table 1. Grid cell modules quantification.**

| Module | ID | spacing | N | pure cells | $\Gamma_1$ (pure) | $\Gamma_2$ (pure) |
|---|---|---|---|---|---|---|
| $S_{59}$ | S1 | $59 \pm 13$ | 140 | 72 | 0.18 (0.60) | 0.29 (0.34) |
| $R^1_{61}$ | R1 day 1 | $61 \pm 17$ | 166 | 93 | 0.23 (0.75) | 0.62 (0.71) |
| $R^1_{58}$ | R1 day 2 | $58 \pm 19$ | 189 | 111 | 0.64 (0.78) | 0.79 (0.83) |
| $R^2_{85}$ | R2 day 1 | $85 \pm 12$ | 168 | 149 | 0.77 (0.77) | 0.89 (0.81) |
| $R^2_{79}$ | R2 day 2 | $79 \pm 18$ | 172 | 152 | 0.85 (0.74) | 0.71 (0.73) |
| $R^3_{121}$ | R3 day 1 | $121 \pm 15$ | 149 | 145 | 0.79 (0.77) | 0.80 (0.86) |
| $R^3_{105}$ | R3 day 2 | $105 \pm 28$ | 183 | 165 | 0.54 (0.45) | 0.80 (0.56) |
| $Q^1_{70}$ | Q1 | $70 \pm 7$ | 97 | 94 | 0.81 (0.74) | 0.74 (0.71) |
| $Q^2_{99}$ | Q2 | $99 \pm 7$ | 66 | 65 | 0.70 (0.62) | 0.64 (0.64) |

long bar in $H_2$ for $R^3_{105}$ is much shorter than the long bars in $H_2$ of the other modules. In general, barcodes that according to the statistical significance test were classified as expressing toroidal topology, have $\Gamma_1$ and $\Gamma_2$ both greater than 0.6 (see Table 1).

When considering all neurons instead of only pure grid cells all datasets that had $\Gamma_1$ and $\Gamma_2$ larger than 0.6 in the latter case maintain this in the former; see Fig 3B. Fig 3E shows the relative difference between $\Gamma$ computed on all grid cells and $\Gamma$ computed on pure cells only. In the case of $S_{59}$ and $R^1_{61}$ both $\Gamma_1$ and $\Gamma_2$ decrease, while an opposite trend is observed for $R^3_{105}$. The decrease in $S_{59}$ and $R^1_{61}$ happens despite the fact that pure grid cells comprise around 50% of all cells in these modules. This is particularly interesting for $R^1_{61}$, for which the subset of pure grid cells had a comparatively high degree of toroidality, but when considering all cells, $\Gamma_1$ drops to 0.25. At least in a simple model of independent Poisson spiking grid cells, Kang et al [14] show that, all things being equal, larger populations should be more likely to express the toroidal topology. So, the results in Fig 3 suggest that the negative effect of head-directional tuning of conjunctive Grid-HD cells on the toroidality in $S_{59}$ and $R^1_{61}$ is so large that it shadows the positive effect of a larger number of neurons.

Thanks to the measure of toroidality, $\Gamma$, we can quantify how the topology of the population activity approaches the toroidality of the full population as the size of the population is changed, and what are, if any, the differences between different modules in this respect. The results are shown for three modules in Fig 4. As expected, the values of $\Gamma_1$ and $\Gamma_2$ increase with the number of neurons, though the dependence varies from module to module: for the module with largest spacing ($R^3_{121}$, Fig 4A), after an initial plateau up to $\sim 50$ neurons, there is a sharp almost linear increase of toroidality with population size. For the module with smallest spacing ($R^1_{58}$, Fig 4G), however, the increase with population size is much smoother and slower, and the module with intermediate spacing is somewhere in between ($R^2_{85}$, Fig 4D); see S1 Fig for the remaining modules and S2 Fig for modules with pure cells only. Fig 4 thus shows that one requires more neurons from a module with small spacing to achieve the same degree of toroidality as the module with larger spacing. This is again counter intuitive, since, as mentioned before, one would expect it to be easier to detect toroidal topology for the module with small spacing because of the presence of more fields in the arena, an expectation that is also confirmed in simulated data [14].

## Spike time jittering destroys toroidal topology at different time scales

To understand the role of temporal order of spikes in the detection of toroidal topology, we jittered experimentally recorded spikes from the population of grid cells that were considered to exhibit the toroidal topology. To this end, we added a random zero-mean Gaussian variable with standard deviation $\Delta t$ to the timing of each spike from each neuron. We then monitored

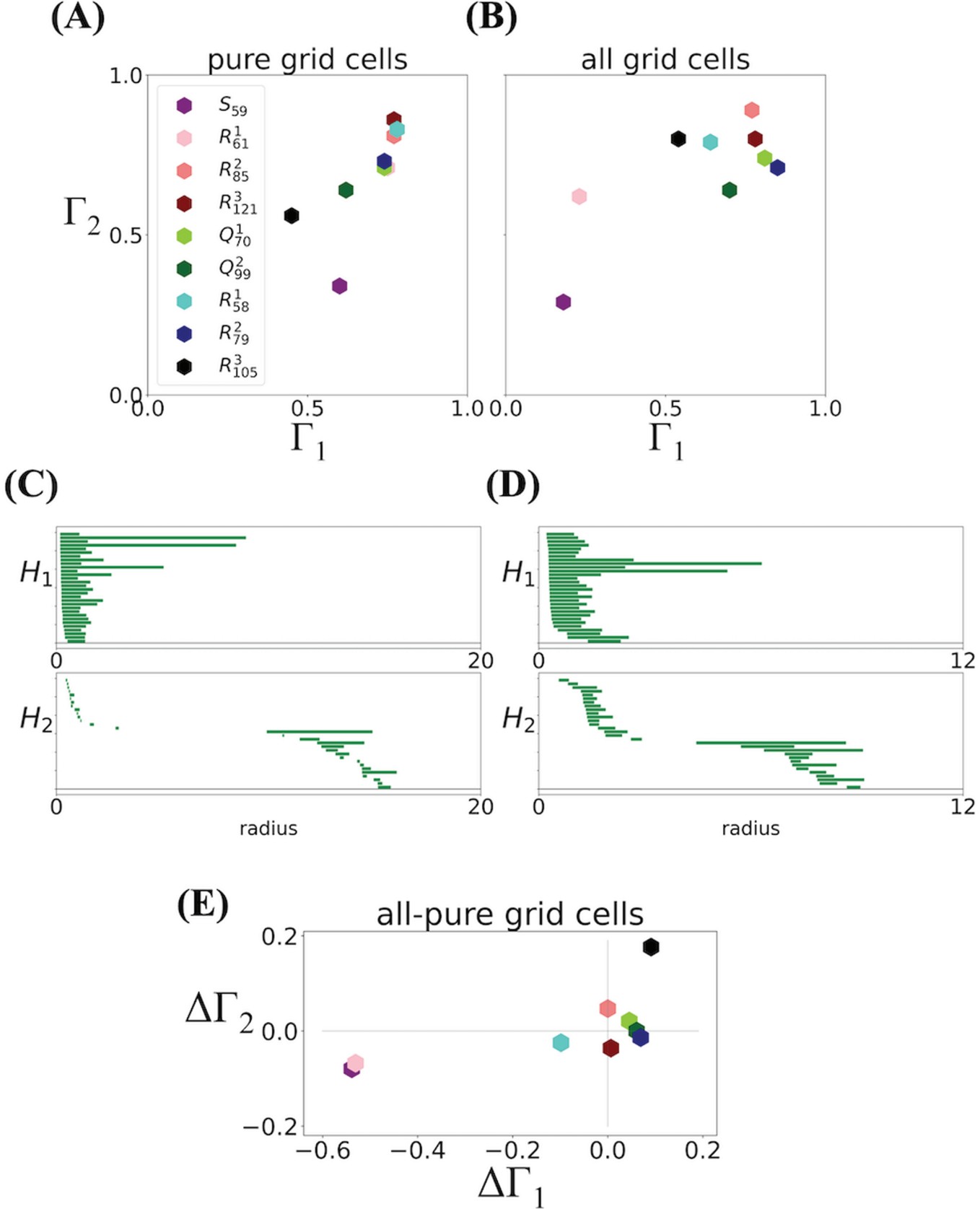

**Fig 3. Degree of toroidality $\Gamma$ in the experimental data.** **(A)** $\Gamma_1$ vs $\Gamma_2$ computed using the subset of pure grid cells in each module. Modules $S_{59}$ (purple) and $R^3_{105}$ (black) have a low degree of toroidality. **(B)** When all neurons are taken into account, module $R^1_{61}$ (pink) exhibits a smaller degree of toroidality,

while the toroidality of $R_{105}^3$ substantially increases.**(C)** Barcode for $R_{105}^3$ with pure cells only, showing a third long bar in $H_1$ and one relatively short bar in $H_2$. **(D)** Barcode for $S_{59}$ with pure cells only, showing multiple bars of comparable length in both $H_1$ and $H_2$. **(E)** The relative difference $\Delta\Gamma = \frac{\Gamma(\text{all}) - \Gamma(\text{pure})}{\Gamma(\text{all}) + \Gamma(\text{pure})}$ between $\Gamma(\text{all})$ computed on all grid cells and $\Gamma(\text{pure})$ computed only on pure cells shows the relative change in the degree of toroidality.

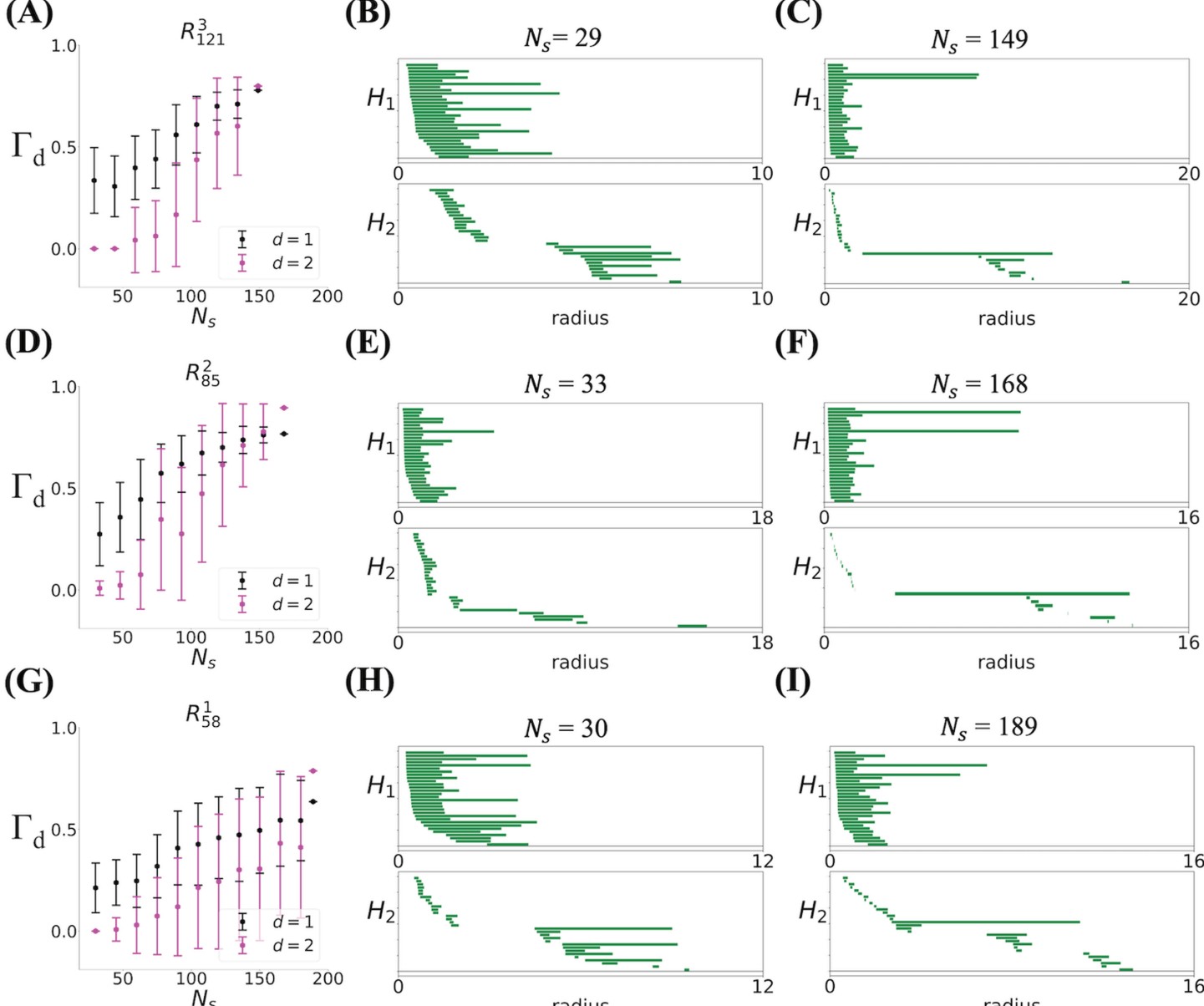

**Fig 4. Degree of toroidality $\Gamma$ smoothly increases as a function of the number of recorded cells.** **(A)** $\Gamma_1$ (black) and $\Gamma_2$ (magenta) computed over a subset of $N_s$ (out of 149) grid cells from module $R_{121}^3$. **(B)** Barcode from module $R_{121}^3$ with $N_s = 29$ has low toroidality. **(C)** Barcode from module $R_{121}^3$ with $N_s = 149$ has high toroidality. **(D)** $\Gamma_1$ and $\Gamma_2$ computed over a subset of $N_s$ (out of 168) grid cells from module $R_{85}^2$. **(E)** Barcode from module $R_{85}^2$ with $N_s = 33$ has low toroidality. **(F)** Barcode from module $R_{85}^2$ with $N_s = 168$ has high toroidality. **(G)** $\Gamma_1$ and $\Gamma_2$ computed over a subset of $N_s$ (out of 189) grid cells from module $R_{58}^1$. **(H)** Barcode from module $R_{58}^1$ with $N_s = 30$ has low toroidality. **(I)** Barcode from module $R_{58}^1$ with $N_s = 189$ has high toroidality. Each point is averaged over 30 different subsets of $N_s$ cells. The last points in each plot are computed on the entire dataset, thus they are single realizations.

how $\Gamma_1$ and $\Gamma_2$ change as a function of $\Delta t$. To calculate $\Gamma_1$ and $\Gamma_2$ for each population, reference barcodes were constructed from the barcodes of the corresponding non-jittered data, as we did in the previous sections; see *Defining the degree of toroidality*. Examples of rate maps from experimental data, corresponding barcodes, and UMAP embedding of population vectors are shown in Fig 5A. After jittering spikes by $\Delta t = 125$ ms (Fig 5B), the ratemaps, barcodes, and the UMAP embedding are shown in Fig 5C. For this value of jitter, the barcodes for this module show little change and the torus persists. Similarly, the degree of toroidality changes only slightly from $\Gamma = (0.79, 0.80)$ to $(0.73, 0.75)$. As shown in Fig 5D, jitters of this size also have little effect on the grid score of neurons in this module.

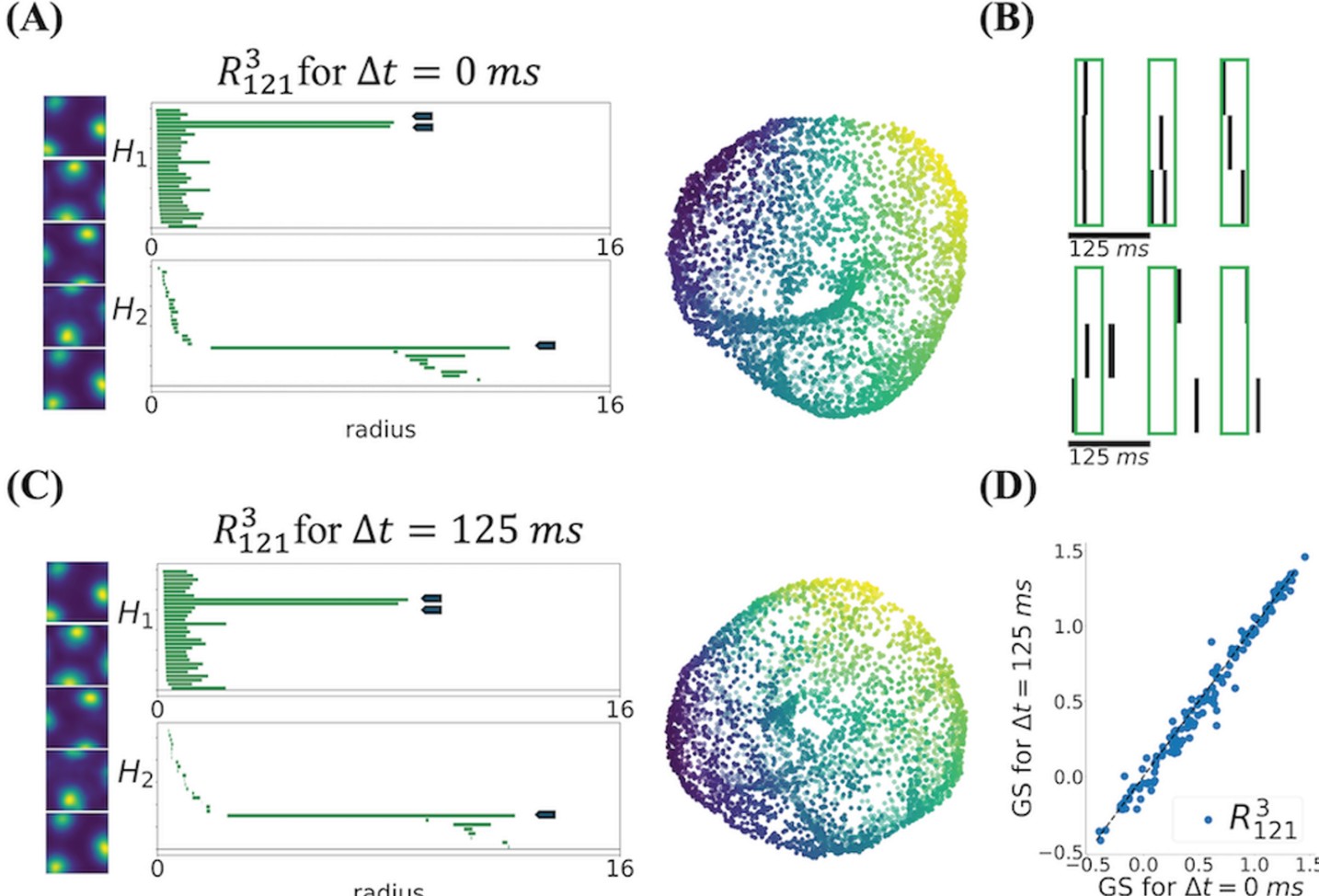

**Fig 5. Jittering the spikes by 125 ms does not destroy the toroidal topology and leaves the grid scores practically unchanged. (A)** Left. Rate maps of 5 representative grid cells from a population of 149 from module $R^3_{121}$ Middle. Barcodes in dimension one ($H_1$) and two ($H_2$) with toroidality $\Gamma = (0.79, 0.80)$. Long, significant bars are indicated by blue arrows. The ordering of the bars along the y-axis is not meaningful. Right. 3-dimensional UMAP embedding of population activity. The color of each point represents the angle along a chosen axis and it is shown only for visualization. **(B)** Spike times from three simultaneously recorded grid cells showing synchronous theta modulation on the top. Spike times were jittered by zero-mean Gaussian numbers with standard deviation $\Delta t = 125$ ms which removed theta correlation, on the bottom. Note that, as a result of jittering, some spikes went out of the depicted range. **(C)** Same as (A) but for jittered spike trains, which yield toroidality $\Gamma = (0.73, 0.75)$, similar to unjittered toroidality. **(D)** The grid score of each cell in module $R^3_{121}$ for the non-jittered and jittered spike trains was similar; $GS = 0.58 \pm 49$ before jittering and $GS = 0.57 \pm 50$ after jittering.

In fact, this pattern was observed for all modules: jitter magnitudes up to a certain level had a minimal impact on toroidal topology. Larger jitter magnitudes, however, eventually cause the degree of toroidality to drop substantially, indicating that the barcodes are no longer consistent with a torus. This is shown in Figs 6 and S3 where a sigmoidal relationship between $\Gamma$ and the size of the jitter $\Delta t$, similar to what was shown in Fig 2 can be observed. This allows us to define, for each module, a critical value $\Delta t_C$, beyond which there is a precipitous loss of toroidality. Specifically, since small values of either $\Gamma_1$ or $\Gamma_2$ indicate lack of a toroidal topology, we define this critical value as the smallest of the inflection points of sigmoidal fits of $\Gamma_1$ and $\Gamma_2$ to $\Delta t$; see *Estimating time scales* in Methods for details.

The critical value $\Delta t_C$ varied systematically from 103 to 484 ms (Table 2), with modules with larger grid spacings losing toroidality at larger temporal jitters than those with smaller spacings. Furthermore, the slope of the sigmoidal fit at the inflection point defining $\Delta t_C$ rapidly increases by increasing spacing. Both these indicate that for small modules, the emerging toroidal topology is much more sensitive to variability in spike times than the larger modules. In other words, given the stochastic nature of the spiking of neurons, a recording from

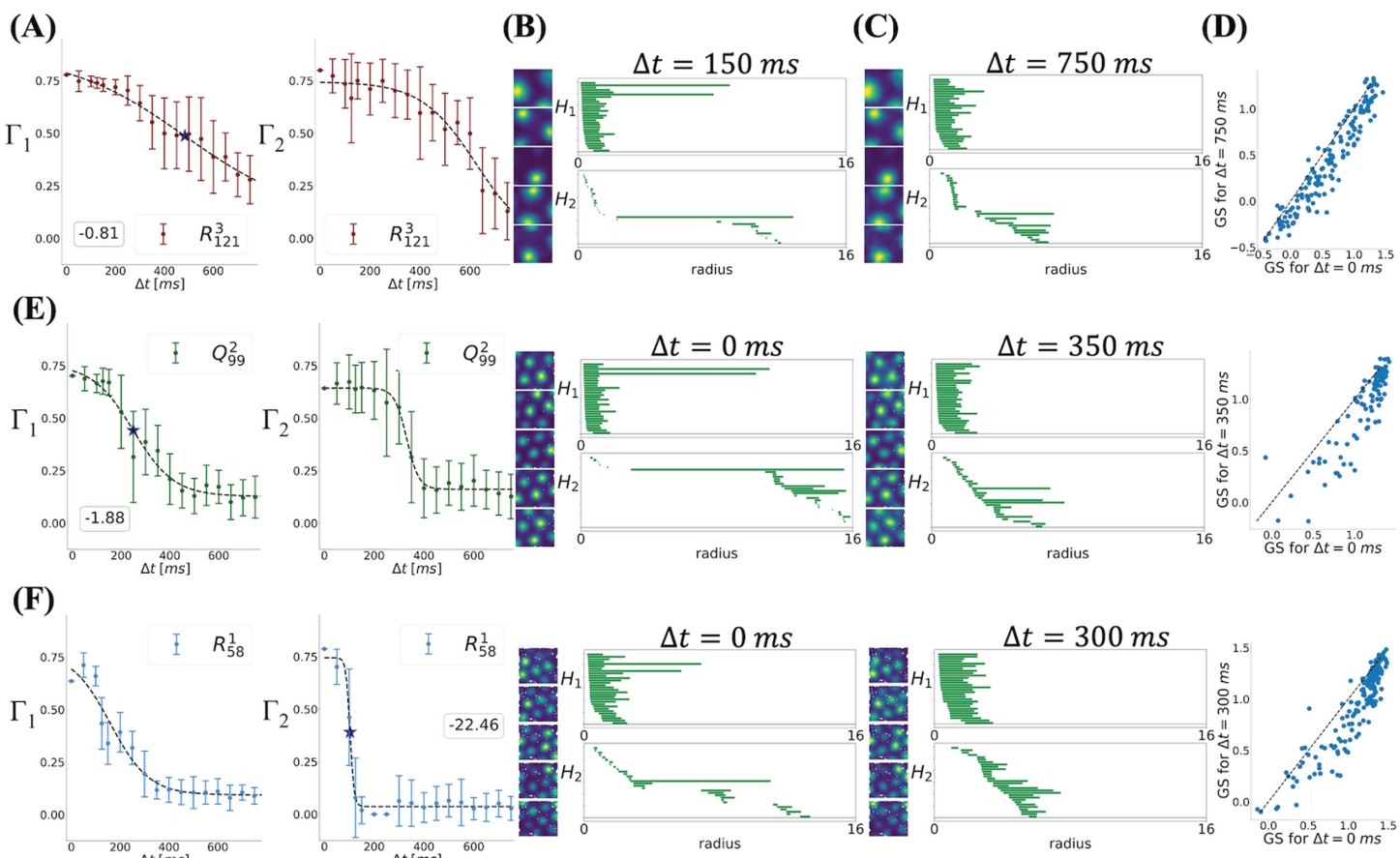

**Fig 6. Toroidality has a sigmoidal dependence on temporal jitter magnitude over a range in which hexagonality is maintained.** (A) Effect of jittering spike times on the degree of toroidal topology of module $R_{121}^3$ ($\Gamma_1$ on the left and $\Gamma_2$ on the right) is sigmoidal. The star shows the value of $\Delta t_C$ (see Methods *Estimating time scales* for details) and the inset shows the slope at the inflection point. (B) Subset of 5 (out of 149) rate maps (left) and the barcode (right) relative to the entire population for a value of jitter smaller than $\Delta t_C$ ($\Delta t = 150$ ms) showing toroidal topology. (C) Subset of 5 (out of 149) rate maps (left) and the barcode (right) relative to the entire population for a value of jitter larger than $\Delta t_C$ ($\Delta t = 750$ ms) which is inconsistent with toroidal topology. (D) Grid scores of the jittered spike trains show little change compared to the unperturbed ones. (E) Same as (A)-(D) for module $Q_{99}^2$. (F) Same as (A)-(D) and (E) for module $R_{58}^1$.

**Table 2. Critical timescale values.**

| Module | $\Delta t_C^{(1)}$ [ms] | $\Delta t_C^{(2)}$ [ms] |
|---|---|---|
| $R_{58}^1$ | 160 | 103 |
| $R_{85}^2$ | 294 | 302 |
| $R_{79}^2$ | 472 | 428 |
| $R_{121}^3$ | 484 | 625 |
| $Q_{70}^1$ | 179 | 203 |
| $Q_{99}^2$ | 250 | 331 |

a smaller module may show toroidal topology, while another recording from the same module may not; this issue will be further discussed in the following sections. Although this could be explained by smaller field size of the smaller modules, one should also consider the fact that smaller modules have more fields in the environment. Taking the limit of very large fields, only one field will be visible in the environment. If the regular organization of the fields plays the key role in the appearance of toroidal topology, toroidality should be absent in such a limiting case [14]; the trend shown by the data, however, is inconsistent with this limit.

The critical jittering time scales of 103 to 484 ms are also much larger than the single neuron integration time constant, but much smaller than the time it takes for the animal to move from one grid field to another; see *Estimating time scales* in Methods. Instead, the time scale of the critical jitter more closely resembles the range associated with the oscillations in the theta (∼125 ms) and eta (∼250 ms) bands [37]. Given the fact that oscillations can help decrease variability in spikes [40,41], it is thus reasonable to assume that they play a role in the appearance and robustness of the toroidal topology.

In order to better understand the roles that these oscillations could play in the appearance of the toroidal topology, we thus simulated a simple population of Poisson-spiking neurons with rates exhibiting hexagonally arranged spatial fields and studied the effect of temporal oscillations modulating these rates.

## Computational model of the effect of oscillations

We consider a population of $N$ Poisson-spiking neurons, indexed by $i = 1, \cdots N$, and each with a firing rate $\lambda_i(\mathbf{r}, t)$ that depends on the position of the animal $\mathbf{r}$ in the 2D box and time $t$:

$$\lambda_i(\mathbf{r}, t) = \left[ \left( \lambda_0 + \sum_k G(|\mathbf{r} - \mathbf{r}_{ik}|) \right) \left( c_1 + c_2 \sum_\mu^m A(\omega_\mu) \cos(2\pi \omega_\mu t) \right) \right]_+ . \tag{3}$$

where $[\cdot]_+$ indicates rectification. Here, the shape of individual fields are modeled by a truncated 2D Gaussian as

$$G(\mathbf{x}) = \frac{G_0}{2\pi\sigma^2} \exp\left( -\frac{\mathbf{x}^2}{2\sigma^2} \right) \left( 1 - \Theta\left[ |\mathbf{x}| - x_0 \right] \right) \tag{4}$$

where $\Theta[\cdots]$ indicates the Heaviside step function. In Eq. (3), $r_{ik}$ is the position of the center of the $k$-th spatial field of the $i$-th neuron. To determine these positions, we first consider a hexagonal lattice with a given spacing in 2D. For each neuron, the lattice is then randomly shifted, and the spatial fields of the neuron are centered on the lattice points. The rate of neurons in Eq. (3) thus exhibits hexagonal spatial regularity imposed by the terms in the first parentheses and temporal oscillations imposed by the terms in the second parentheses. The

parameters $c_1$ and $c_2$ in Eq. (3) determine the oscillatory modulations: for $c_1 = 1$ and $c_2 = 0$ there are no oscillatory modulations, while for $c_1 = 0$ and $c_2 \neq 0$, spatial ratemaps are modulated temporally by synchronous oscillations. Unless otherwise stated, in the simulations that follow, we set $\lambda_0 = 0.05$, $G_0 = 1.5$, $x_0 = 0.4$, and $N = 75$; see also *Parameters of the oscillations* in Methods. The simulations were run over half the trajectory of the first recording (day 1) of rat R; the time period was divided into bins of $\delta t = 10$ ms and the number of spikes of neuron $i$ in a bin $[t, t + \delta t]$ was generated from a Poisson distribution with mean $\lambda_i(\mathbf{r}(t), t)\delta t$.

In Fig 7 we show an example of simulating the model, demonstrating that in the presence of oscillations, the model exhibits toroidal topology. The simulated population had a spacing similar to $R^2_{85}$ and we set $\sigma = 0.12$ to obtain field sizes comparable to the same module. The power spectrum of the neural activity in these simulations was chosen such that $A(\omega) \propto \omega^{-1/2}$ but with stronger amplitude oscillations at frequencies 4 Hz and 8 Hz added to it; see *Parameters of the oscillations* in Methods for details. The rate map and the power-spectrum density of the spike train of one cell for these choices of parameters are shown in Fig 7A and 7B. The barcodes and UMAP embedding in Fig 7C and 7D show the presence of a toroidal topology.

To better understand how the behavior of the model depends on the parameters, in Fig 8 we show how toroidality changes as a function of the fields size, $\sigma$. Fig 8A and 8B show this when the toroidality is calculated with respect to the idealized torus constructed from a real dataset with similar spacing (dataset $R^2_{85}$). We find that when the field sizes (controlled by $\sigma$) are large enough, all the realizations exhibit a large degree of toroidality. Importantly, however, when oscillations are removed, the barcodes become very different from the toroidal topology of the real data (lower $\Gamma_1$ and $\Gamma_2$ averaged over realizations). At fixed $\sigma$, we also observe a larger variability in the degree of toroidality from one realization to another in the absence of oscillations compared to when oscillations are present.

Since the definition of toroidality depends on the choice of the idealized reference torus, we also measured toroidality when the reference barcode in Eq. (1) is constructed from the simulation itself, by keeping the longest bar in $H_1$, setting the second longest bar to the length of this longest bar, and also keeping the longest bar in $H_2$. The lengths of all other bars in a given dimension were set to the smallest bar length in that dimension. We denote the

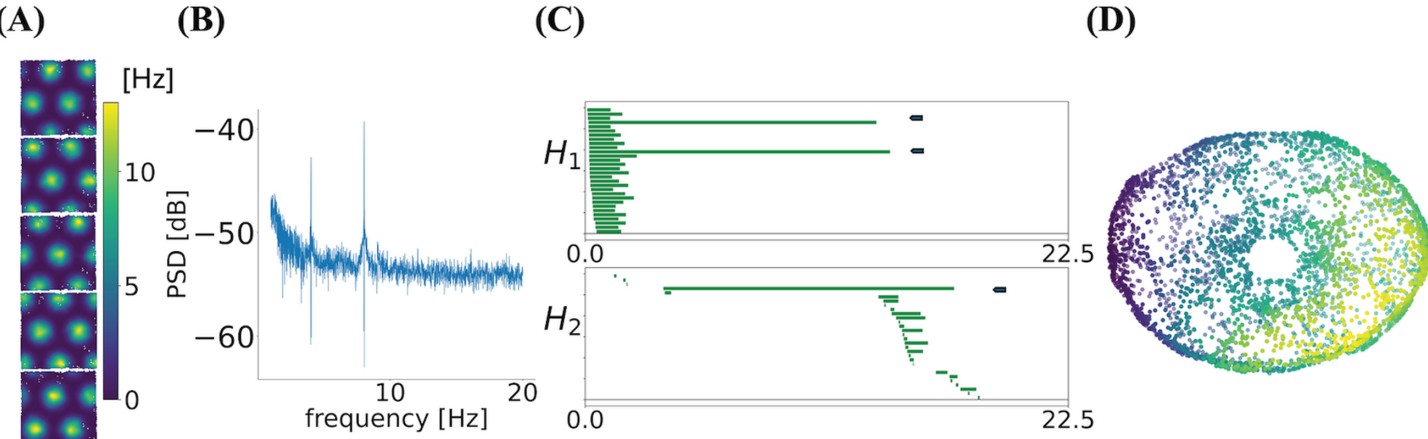

**(A)** **(B)** **(C)** **(D)**

**Fig 7. Simulation of a grid cell module from Eq. (3).** **(A)** Subset of 5 (out of 75) rate maps with spacing similar to $R^2_{85}$. **(B)** Power spectral density of a single cell simulated spike train showing peaks in the eta (4 Hz) and theta (8 Hz) frequency bands, as constructed in Eq. (3). **(C)** Barcode from the persistent homology of a population of 75 simulated grid cells show clear toroidal topology. **(D)** The 3D UMAP embedding of the population activity shows a 3-dimensional torus. The color of each point represents the angle along a chosen axis and it is shown only for visualization.

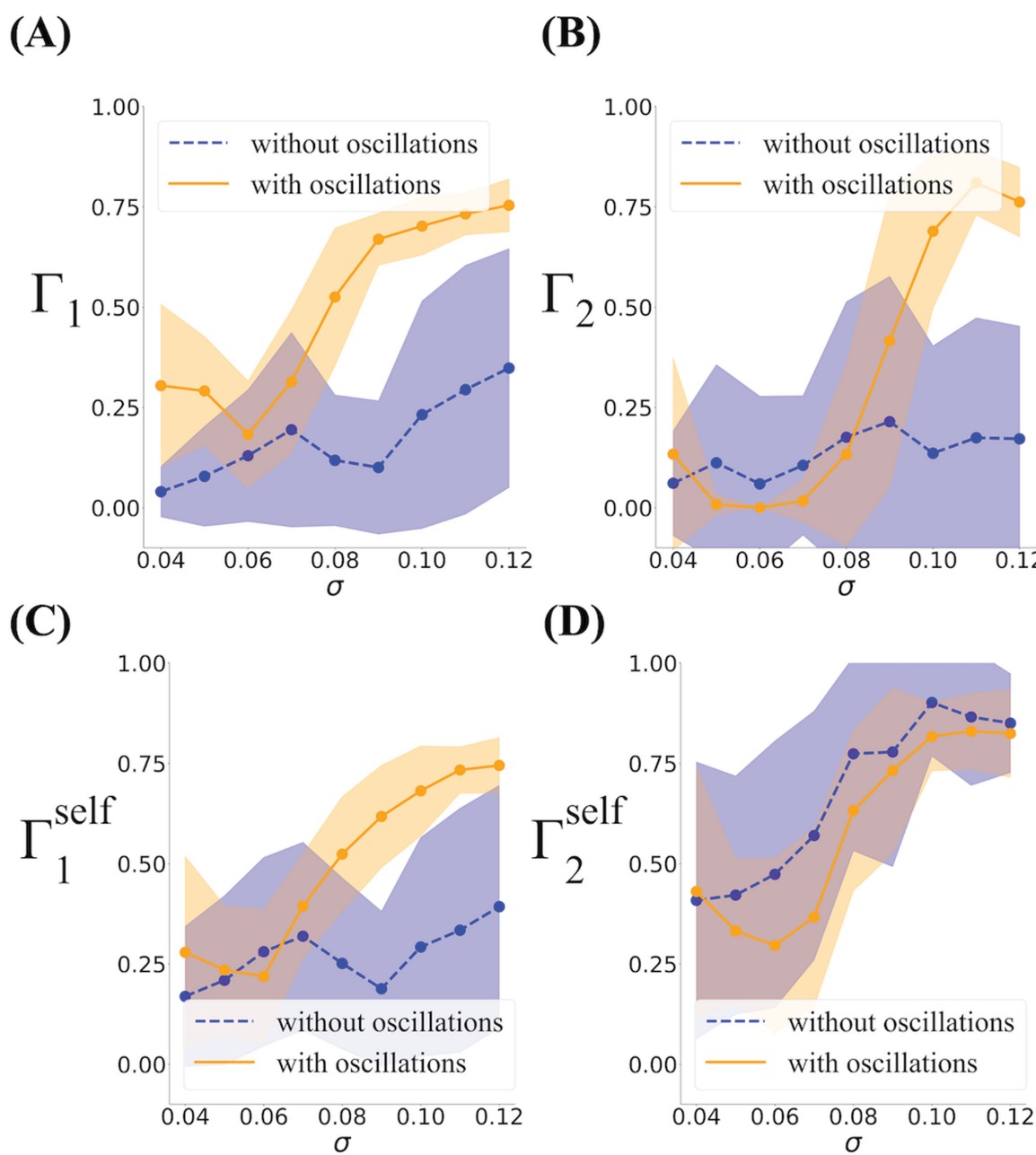

**Fig 8. The dependence of the degree of toroidality on the field size $\sigma$ with and without oscillations. (A, B)** The barcodes of the simulated populations modulated by oscillations are similar to the associated experimental module for large enough values of $\sigma$, averaged over 20 realizations (shaded area indicates standard deviation). The parameters for the oscillations are as described in Methods. Removing the oscillations dramatically reduces this similarity and increases the variability both in $\Gamma_1$ (A) and $\Gamma_2$ (B). **(C, D)** Similarity measures $\Gamma_1^{self}$ and $\Gamma_2^{self}$ between the barcodes from the simulations and a reference barcode produced from the simulated population with two long bars in $H_1$ and one long bar in $H_2$, as descried in text.

resulting similarity measure by $\Gamma^{self} = (\Gamma_1^{self}, \Gamma_2^{self})$ and show the dependence of this measure on $\sigma$ in Fig 8C and 8D. Just like $\Gamma_1$, for simulations without oscillations, the mean value of $\Gamma_1^{self}$ is substantially lower than one and its variability is high (Fig 8C). As opposed to $\Gamma_2$, however,

large values of $\Gamma_2^{self}$ were indeed achieved for a sufficiently large $\sigma$ both in the presence and in the absence of oscillations (Fig 8D). We remind that toroidal topology requires both components to be simultaneously close to one. It is thus possible but improbable that the simulations in Fig 8 exhibit toroidal topology; some may do given the high variability from simulation to simulation. In Fig 9A-C the barcodes of one such rare run is compared to those of the real data and the simulations in the presence of oscillations. In the simulations with oscillations (Fig 9A) and in the real data (Fig 9B), the long bars in $H_1$ nd $H_2$ all co-exist over a range of scales. However, even in the exceptional examples where both $\Gamma_1^{self}$ and $\Gamma_2^{self}$ are large in the absence of oscillations (Fig 9C), the long bar in $H_2$ appears when those in $H_1$ have disappeared. In fact, as Fig 9D shows, this difference is a general feature of the barcodes that the algorithm finds in these simulations in the absence of oscillations.

The inclusion of two dominant oscillatory components at 4 and 8 Hz is motivated by previous experimental results on theta and eta band modulations. In Fig 10A we show how including only one dominant frequency, and varying the frequency of this oscillation changes the results. Here, we performed multiple simulations with the same parameters as in Fig 7, but by removing the oscillation at eta or theta (e.g., setting $A(\omega_\mu) = 0$ for the eta oscillation) and

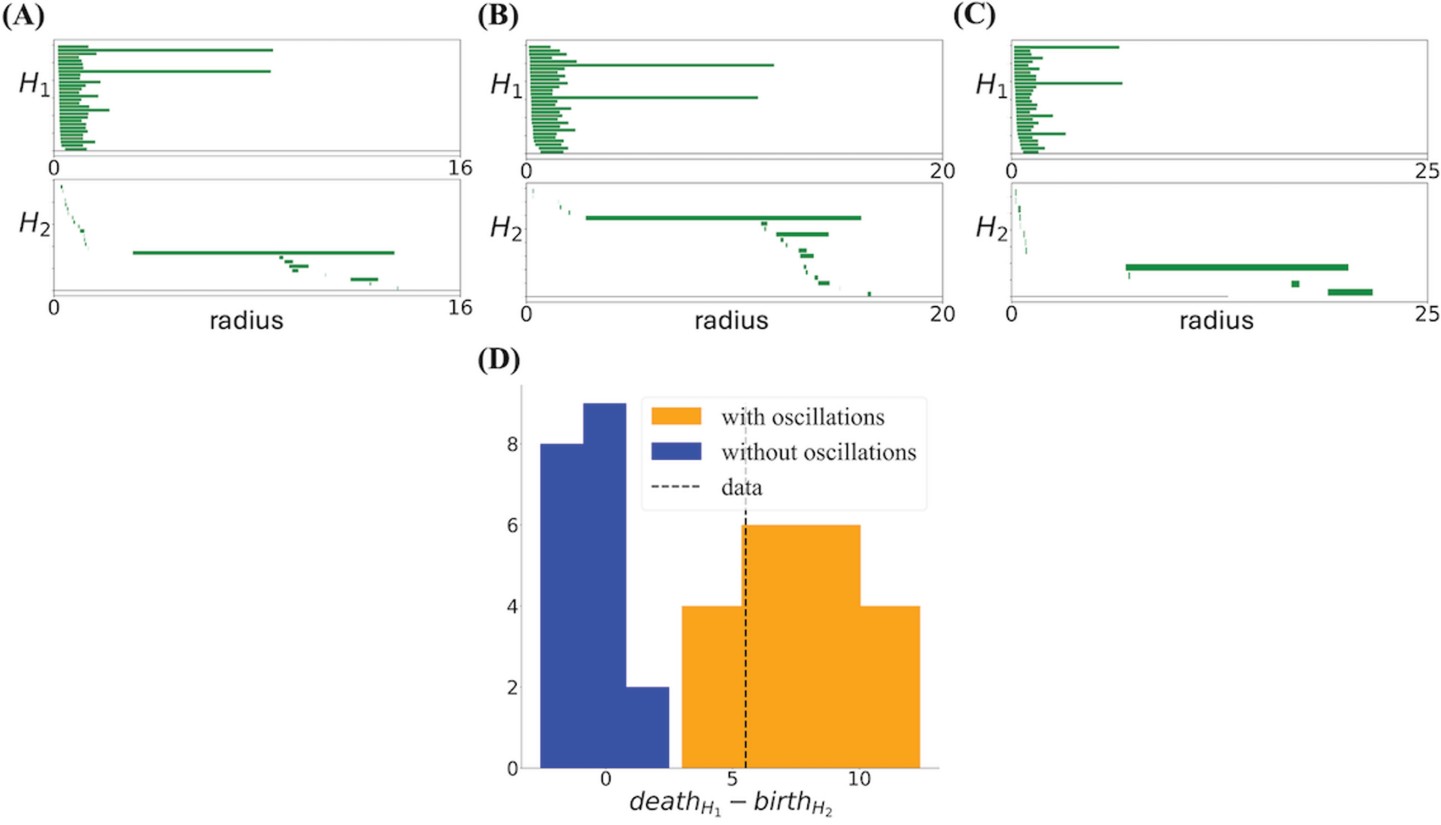

**Fig 9. Barcodes for the simulated population with oscillations are similar to data.** Barcodes from experimental data **(A)**, simulation with oscillatory modulation **(B)** and simulation without **(C)**. The barcodes from simulations with oscillations (B) look very similar to the data barcode (A), as quantified by $\Gamma$. The parameters of the oscillations are as described in Methods. When oscillations are removed (C), toroidal topology can still be present, but the barcodes are far from the data. **(D)** The histogram shows that the difference between the death radius of the longest bar in $H_1$ and the birth radius of the longest bar in $H_2$ takes a large positive value in the data and in the simulation with oscillations, while it is much closer to zero in the simulation without oscillations. The long bar in $H_2$ is consistently born at the same time as the longest bars in $H_1$ die. The histogram is from 20 realizations in each case.

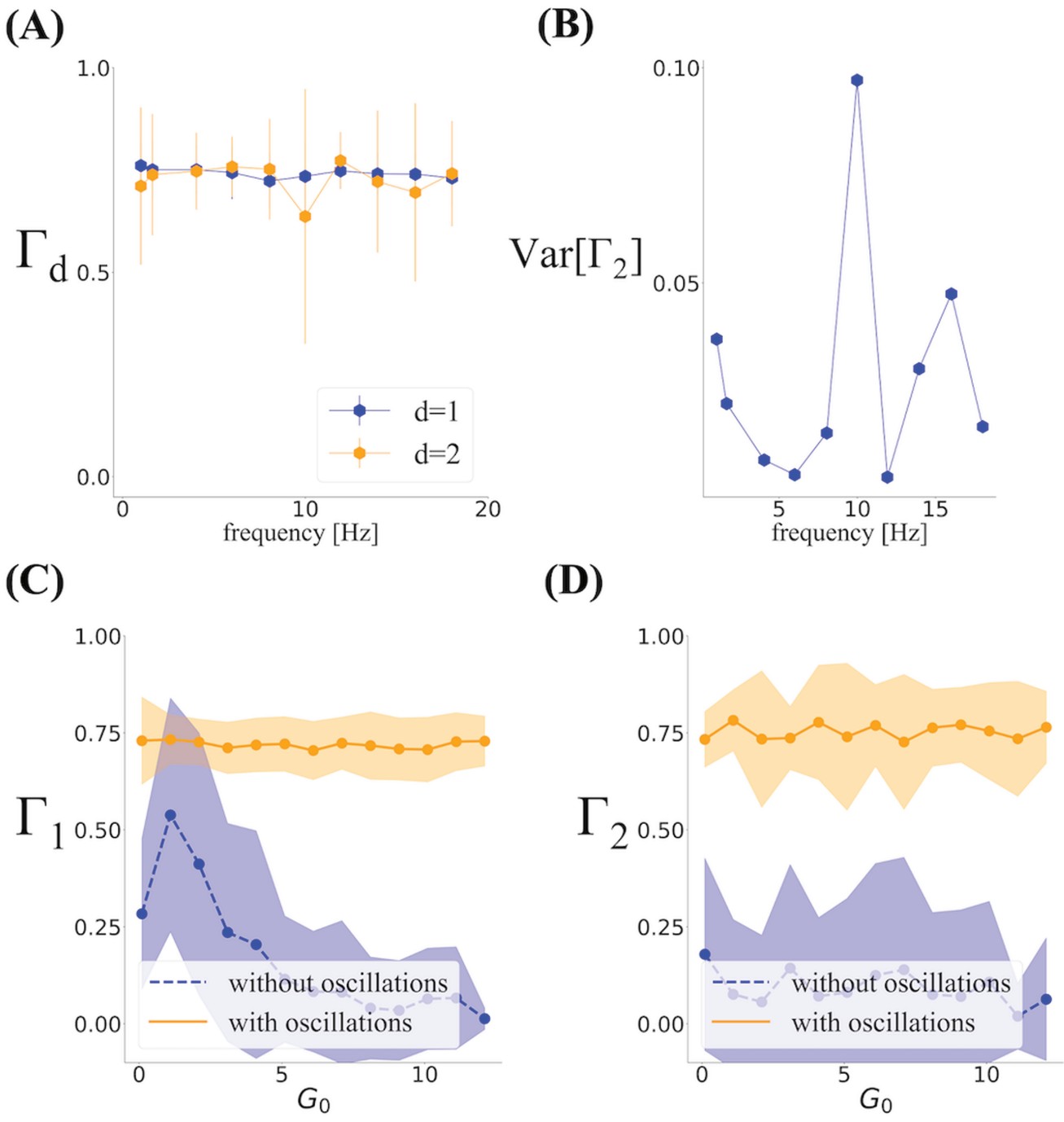

**Fig 10. Dependence of $\Gamma$ on firing rate and oscillation frequency.** **(A)** The simulation with one single oscillator shows that $\Gamma$ is not dependent on the oscillator frequency in a module with spacing similar to $R_{85}^2$. Except for the choice of the dominant frequency described in the text, the parameters for the oscillations are as described in Methods.**(B)** The variance of $\Gamma_2$ shows a minimum at the eta and theta time scales. Simulation parameters are the same as Fig 7, removing the oscillation at eta, and varying the frequency of the remaining one. The errors bars are over 30 realizations. **(C, D)** Toroidality increases when cells are modulated by oscillations for a large span of $G_0$ values both in $\Gamma_1$ and $\Gamma_2$. The averages and errorbars for each value of $G_0$ are over 20 realization of the simulations.

varying the frequency of the remaining dominant oscillation. It is evident that the degree of toroidality averaged over multiple realizations takes a large value, even in the presence of only one oscillator, regardless of its frequency. Importantly, however, we find that how consistently one obtains a large degree of toroidality does depend on the chosen frequency. This is reflected in the fact that the variability of $\Gamma_2$ from one simulation to another shows a minimum in the eta to theta range and their harmonics (Fig 10B). Similar results are obtained when we consider the effect of oscillations in simulations with smaller spacing, as shown in S4 Fig.

In our simulations so far, model neurons had idealized hexagonal spatial tuning. Real neurons, however, exhibit irregularities that can significantly decrease the chance of detecting a toroidal topology. This raises the question of how much oscillations, or lack of them, contribute to the toroidal topology when compared to the degree of hexagonal organization. In Fig 11 we thus show the effect of perturbing the position of the grid field centers. The results show that, in the presence of oscillations, toroidal topology is robustly present up to a displacement of around 10-15 % of grid spacing, where it suddenly drops. As can be seen from the rate maps in Fig 11, this is a quite substantial perturbation of the hexagonal regularity of the grid cells. In the same way, even when the hexagonal regularity is replaced by a square lattice, barcodes similar to the real data can be observed, and the effect of oscillations is similar to the hexagonal lattice (S5 Fig). In summary, at least when oscillations are present, lack of idealized hexagonal field arrangements in the simulations has a weak effect on obtaining barcodes similar to the experimental data.

The results so far indicate that oscillations contribute to the appearance of toroidal topology. This effect is likely achieved through the influence of the oscillations on the variability of neural spiking at the single neuron and population levels. To better understand this influence, it is necessary to look more closely at how persistent homology detects structure in data. To obtain barcodes consistent with toroidal topology in real data, persistent homology is applied to a subset of population vectors with the highest mean activity [12]; see Experimental Procedures and Preprocessing. As shown in S1 Appendix, this choice has a significant effect on the detection of toroidal topology: applying persistent homology on randomly chosen time points from the real data and simulations with oscillations drastically decreases the degree of toroidality. On the other hand, applying persistent homology to these randomly chosen population vectors in Poisson simulations without oscillations has the opposite effect. This means that patterns in population activity that form the toroidal topology in real data are present at high activity time points. They are also not the same as those patterns that arise from overlapping hexagonally organized fields present in simulations without oscillations. The difference is also not due only to the increased firing rate of the sampled population vectors. In fact, as shown in Fig 10C and 10D, increasing the mean firing rate in simulations by increasing $G_0$ in Eq. (4) does not generally change the picture on the effect of oscillations in simulations. Although increasing $G_0$ appears to initially increase $\Gamma_1$ in the absence of oscillations, this is still small and the variability is considerable, even when the number of cells is doubled; see also S6 Fig and S1 Appendix.

The simulation results reported in this section confirm the hypothesis we made based on the jittering analysis: that there is a crucial role for oscillations in the high toroidality found in experimental data. However, our simple simulated network differs from the real network in many respects. For example, the power spectral density of real data often shows peaks that have a non-zero width. Furthermore, they do not only modulate the rates but also influence the timing of spikes and the order at which neurons spike [42,43]. Most importantly, contrary to the simulated data, where the field size, oscillatory structure, firing rates and other relevant parameters can be changed independently of one another, in experimental data, oscillations

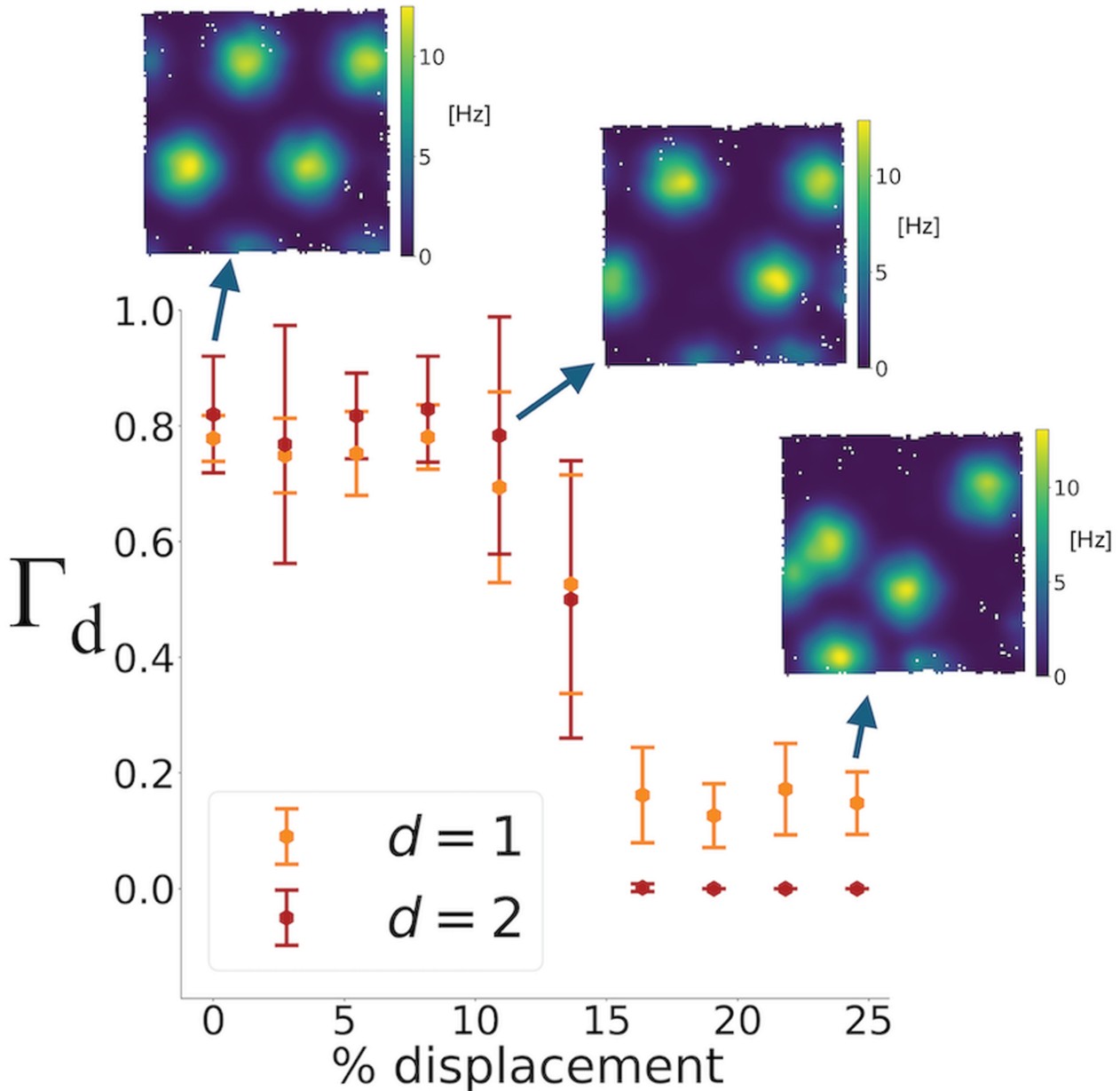

**Fig 11. The dependence of $\Gamma$ on grid field displacement.** The degree of toroidal topology of the simulated populations modulated by oscillations is refractory to grid field center displacement up to around 10-15% of the grid spacing, where it drops suddenly. The arrows show a rate map example for the following levels of relative displacement: 0%, 12% and 24%. The mean and error bars are from 20 realizations of the simulations with the parameters for the oscillations described in Methods.

are correlated with all these other features [43–45]. As a result, to test the role that oscillations play on toroidality in real data and to quantitatively assess what oscillation features, e.g. number of prominent oscillators, their frequency bands, their amplitude, contribute to this role, we again turn to analyzing the experimental data.

### Increased eta-to-theta oscillation power is predictive of larger critical jitter size

We calculated the power spectral densities of grid cells recorded in the experimental data. In Fig 12, we plot the average power spectral densities of neurons from the same module, normalized by the total power in the range of 0.1 – 500 Hz. Because these spectra are known to be influenced by the animal's movement, we plot the spectra for modules recorded from the same animal in the same recording session together in different panels.

The first noticeable observation is that the power spectrum not only exhibited a peak at the ~8 Hz theta rhythm, but also at a lower frequency, the ~4 Hz eta rhythm (Fig 12), similar to what has recently been discovered in the hippocampus [37]. These oscillations were observed in all modules regardless of whether they exhibit a large degree of toroidality or not. However, when we plot the degree of toroidality as a function of the ratio $A_\eta/A_\theta$, as shown in Fig 13, an interesting pattern is noticed: although modules with the larger $A_\eta/A_\theta$ had a large degree of toroidality, modules with $A_\eta/A_\theta$ towards the lower end showed both high and low toroidality. The ratio $A_\eta/A_\theta$ also correlated with the grid spacing.

In other words, datasets containing grid cells with similar small spacing, that is $R_{58}^1$, $R_{61}^1$– the same module recorded on two separate days, thus slightly different estimated spacing– and $S_{59}$, showed the smallest $A_\eta/A_\theta$. Recall that, as shown in Fig 3, these are precisely the

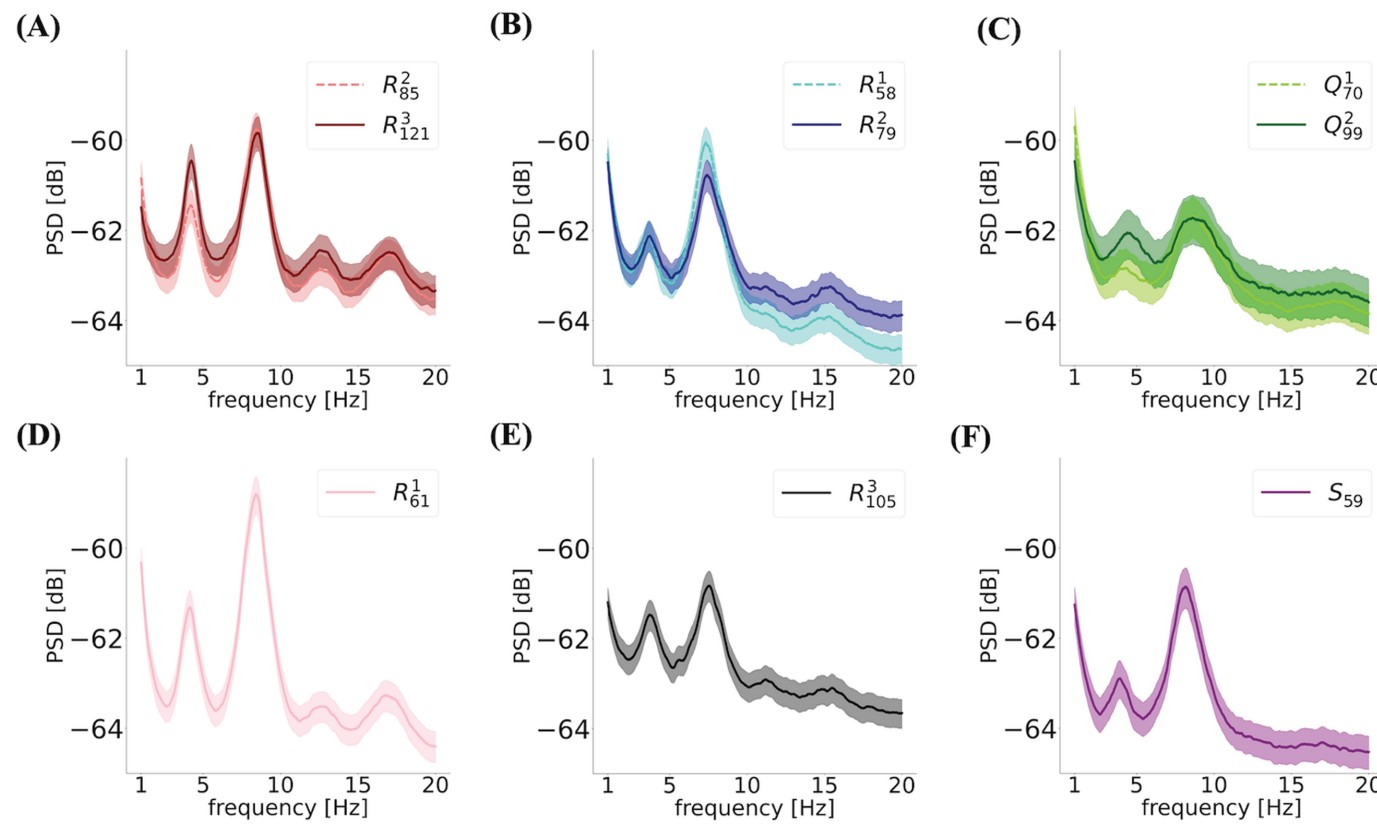

**Fig 12. Firing of grid cells are modulated at eta and theta bands.** The power spectra of the neurons' spike counts from different modules averaged over all neurons (full curves) and corresponding s.e.m. (shaded area) for **(A)**-**(C)** groups of cells recorded from the same animal on the same day that did exhibit high toroidality. **(D)**-**(F)** the three modules that did not show high toroidality when all cells were considered. Modules in panels (A) and (D) and in panels (B) and (E) were simultaneously recorded.

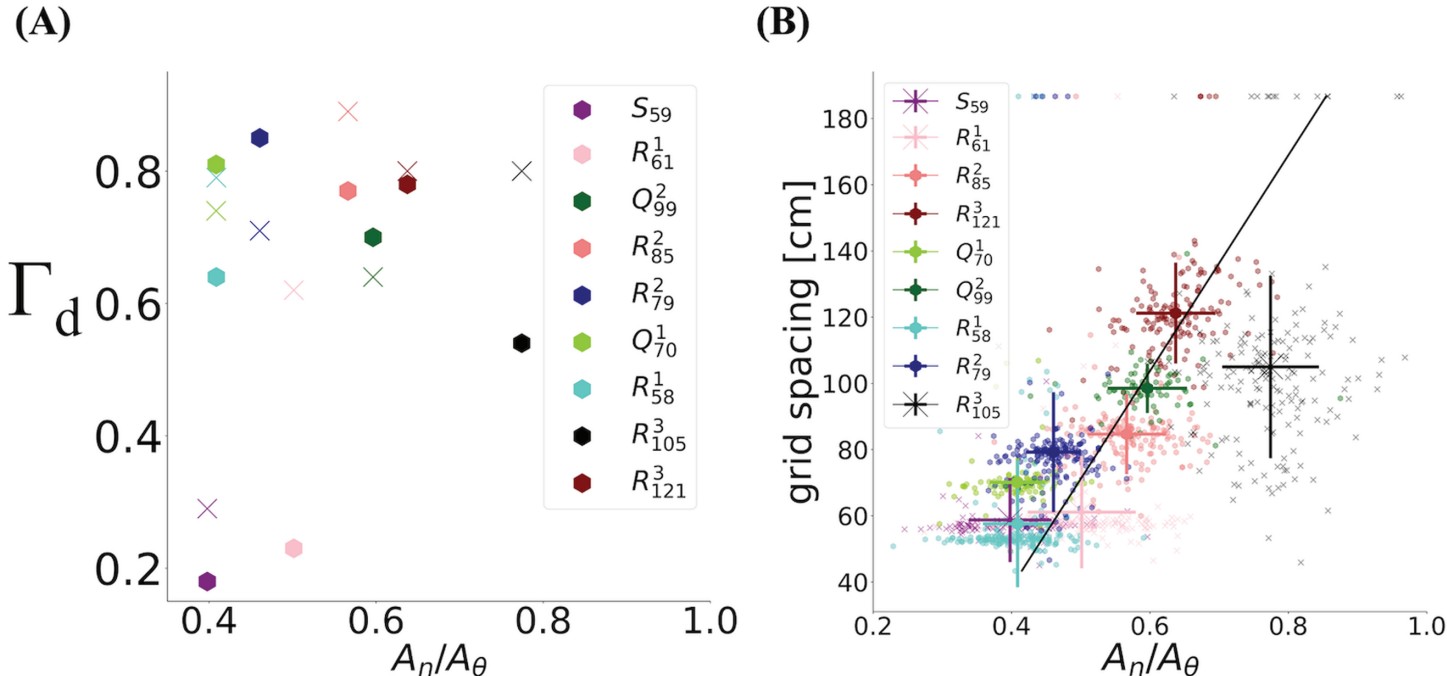

**Fig 13. Dependence of $\Gamma$ and spacing to eta-to-theta power. (A)** While for small $A_\eta/A_\theta$ some modules show high toroidality, while others do not, for all modules that do show large toroidality, there is little dependence of $\Gamma_1$ (hexagon) and $\Gamma_2$ (cross) on eta-to-theta power ratio. **(B)** Grid spacing correlated with eta-to-theta (correlation 0.62, p-value < 0.05). The one outlier $R_{105}^3$ and the samples represented by the cross do not exhibit large toroidality.

same datasets for which either a large degree of toroidality was never observed ($S_{59}$) or only when pure grid cells were considered ($R_{61}^1$), and in only one case ($R_{58}^1$) it was observed with all cells included. Since these modules had similar spacing, this means that the lower $A_\eta/A_\theta$ leads to less robust tori, in the sense that depending on e.g. the composition of the cell population, or from one recording of the same module to another, it can be present or not.

To further test this conclusion, we return to the jittering results and the critical time scales $\Delta t_C$. In Fig 14 we plot $\Delta t_C^{(1)}$ (panel A), $\Delta t_C^{(2)}$ (panel B) and their minimum for a single module $\Delta t_C$ (panel C) as a function of $A_\eta/A_\theta$. This ratio showed a positive correlation with the critical spike-time jitter $\Delta t_C^{(d)}$ that destroys the toroidal topology in dimension d, as well as with $\Delta t_C$. Moreover, comparing the ratio $A_\eta/A_\theta$ between simultaneously recorded modules, we find that this ratio is significantly lower for modules with smaller $\Delta t_C$ compared to ones with larger.

We also tested this relationship for $A_\eta$ and $A_\theta$ independently, but no correlation with the critical jitter size (S7 Fig) was observed. A similar analysis was performed on other spectral components, such as the delta-to-theta and delta-to-eta power ratios. However, the eta-to-theta power ratio showed the greatest correlations, as shown in S7 Fig.

As mentioned above, in the case of small modules, the appearance of the toroidal topology was not a robust phenomenon. If higher $A_\eta/A_\theta$ is the cause here, then simulating a population with similar grid spacing but larger eta-to-theta ratio than in the experimental data should lead to more robust tori. This is indeed the case, as demonstrated in Fig 15. Since the barcodes of the real data from modules with small spacing (< 70 cm) don't show a sufficiently clear torus, we compute the reference torus from the simulation itself, as discussed previously. As can be seen in Fig 15, in addition to having a generally lower toroidality, in simulations

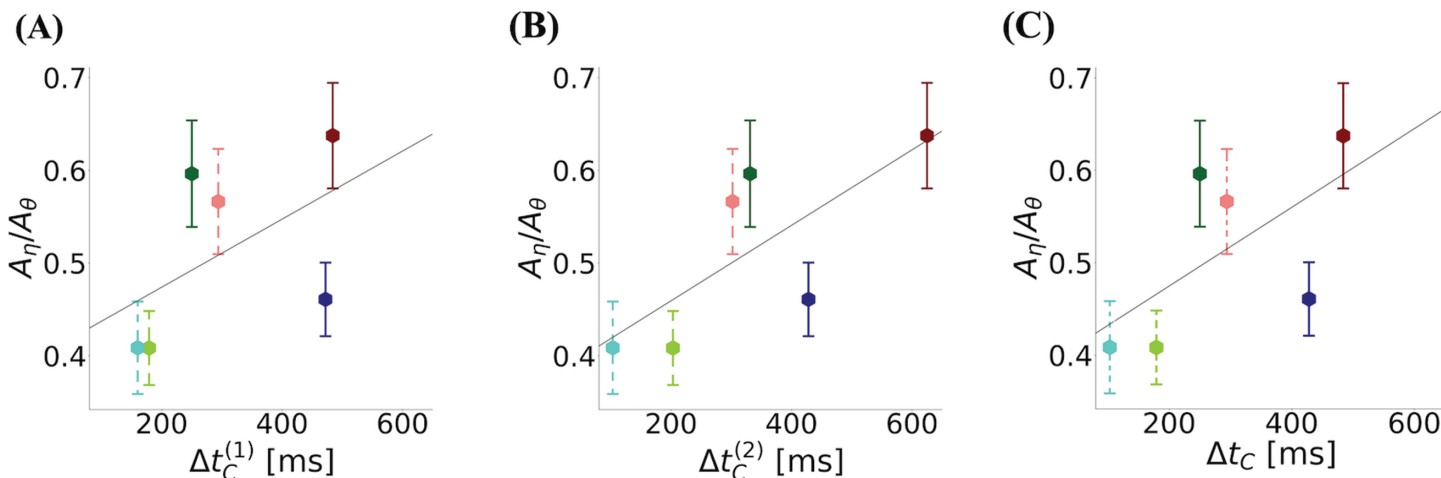

**Fig 14. Eta-to-theta power ratio shows a positive correlation trend with the critical jitter size.** The relationship of **(A)** $\Delta t_C^{(1)}$, **(B)** $\Delta t_C^{(2)}$ and **(C)** $\Delta t_C$ with $A_\eta/A_\theta$. Populations recorded simultaneously are shown with the same color shade, one with full lines, the other with dashed lines. The mean and errorbars are calculated over the neurons from each module. The lines are linear regressions between these means and the corresponding x-axes in each panel ((A) slope = $3.7 \times 10^{-4}$, $r^2 = 0.27$, (B) slope = $4.1 \times 10^{-4}$, $r^2 = 0.55$, (C) slope = $4.3 \times 10^{-4}$, $r^2 = 0.38$). For each pair of simultaneously recoded modules the mean $A_\eta/A_\theta$ was larger in the module with larger $\Delta t_C$ compared to the one with smaller $\Delta t_C$ (p-value<0.001); the same holds for $\Delta t_C^{(1)}$ and $\Delta t_C^{(2)}$.

without oscillations, any potential long bars in $H_1$ exists at scales different from that of $H_2$, making these barcodes very different from those of the experimental data.

## Summary

In this paper, by quantifying toroidal topology using a new measure, $\Gamma$, we first confirmed the presence of a large degree of toroidality in populations of grid cells. The measure defined here, provides a quantitative estimate of the similarity between barcodes, and their associated topological structure. By considering a proper user-defined reference topology, this measure can be applied and extended to arbitrary topologies (e.g. circle, sphere), providing information about underlying mechanisms that is unavailable in significance testing approaches.

To gain insights into what temporal aspects of the neural spike trains underlie this topology, we added temporal jitters to the experimentally recorded spike times. This revealed a sigmoidal dependence between $\Gamma$ and the size of the temporal jitter: temporal jitters below a critical value, ranging from 103 ms to 484 ms depending on the module, had minimal impact, but larger jitters destroyed the toroidal topology. Interestingly, for a range of jitters above the critical value, the hexagonal regularity in the organization of the fields did not change much, thus showing that the regularity was not sufficient for producing toroidal topology.

The critical jittering time scales we found were larger than the single-neuron integration time (10–20 ms) and shorter than the rat's travel time through neighboring grid fields (4–5 s). On the other hand, these critical jittering time scales were much closer to the periods of eta and theta oscillations. Such oscillations are found *in vivo* under various conditions, ranging from delta oscillations during sleep to theta oscillations during behavior [36,37]. We therefore simulated populations of rate-modulated Poisson spiking neurons. The rate was modulated by two parameters: a spatial parameter that enforces the hexagonality of the firing, similar to the experimental data, and a temporal parameter consisting of oscillations with dominant theta and eta components. We found that without the oscillatory components, the barcodes from such populations substantially differed from real data, but adding oscillations changed

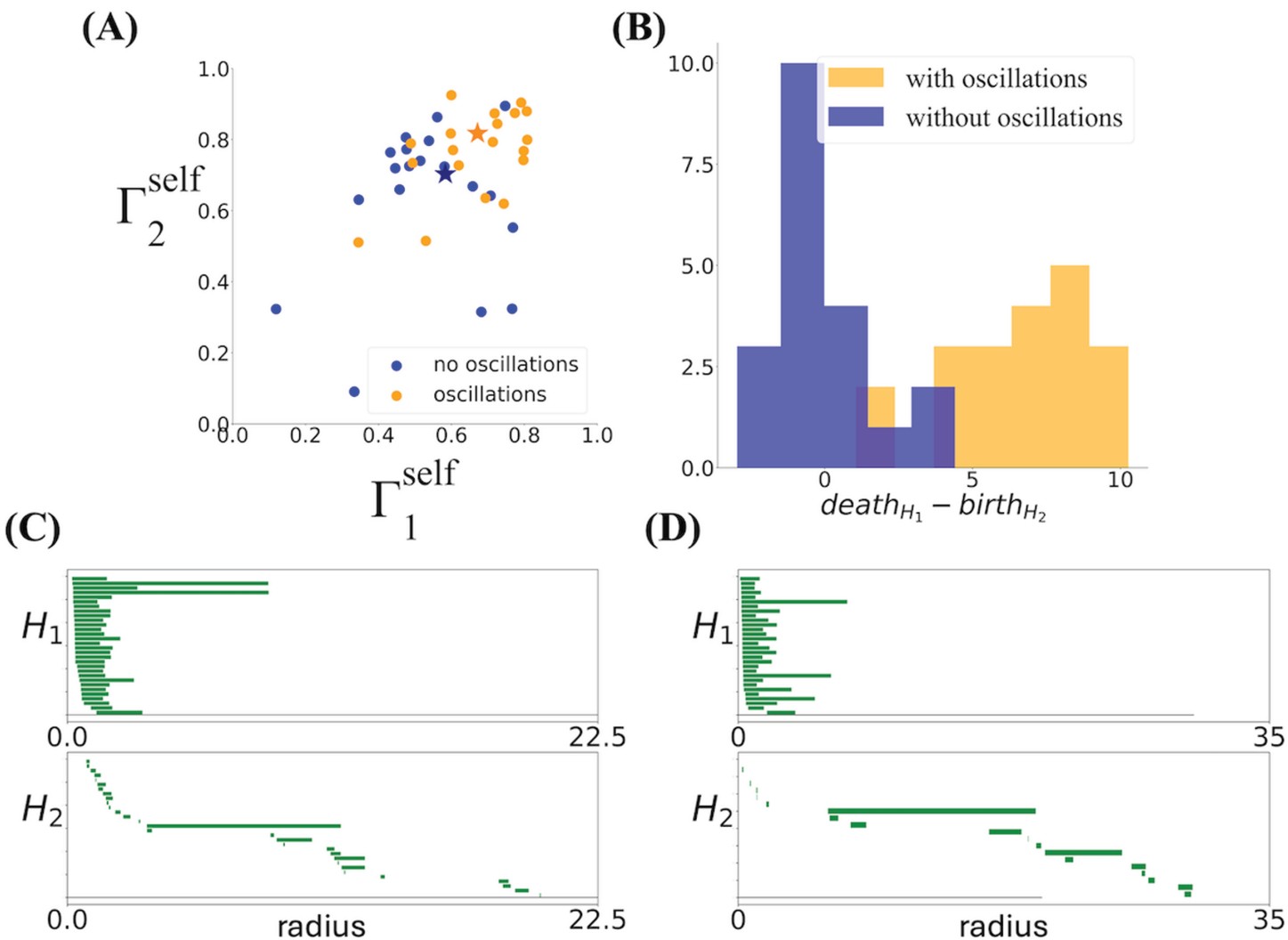

**Fig 15. Toroidal topology is more robust for small spacing when large eta-to-theta oscillations are included.** **(A)** The variability of $(\Gamma_1, \Gamma_2)$ is reduced when eta and theta oscillations are included ($A\eta/A_\theta = 1.1$) in the simulation with the experimental trajectory of the rat from day 2, and the simulated population shows toroidal topology more consistently over 20 different realizations. **(B)** The difference between the death radius of the longest bar in $H_1$ and the birth radius of the longest bar in $H_2$ takes a large positive value in the data and in the simulation with oscillations, while it is much closer to zero in the simulation without oscillations. **(C)** Example barcodes with oscillations and without them **(D)**, corresponding to the simulations indicated by orange and blue stars in (A), respectively.

this. The effect was also present with only one dominant oscillation with a frequency outside the eta or theta bands, but variability in the degree of toroidality from one run of the simulation to another was comparatively smaller in these bands. The high degree of toroidality that we observed in the presence of oscillations was also found to be insensitive to perturbing the position of the fields by up to 10–15%, perturbation which appreciably changes the regularity of the fields. This suggests that the hexagonal regularity may not be a necessary factor in the high toroidality that we obtained for the experimental data.

Given the results of the jittering analysis and the simulations, we thus hypothesized that oscillatory modulations of the firing rates are a crucial factor in the appearance and robustness of toroidal topology in the real data. We tested this hypothesis directly on experimental data. We computed the power spectra of the grid cells spike counts measured during behavior.

This revealed a large power not only at theta (8 Hz) but also at low-frequency delta (0.1–2 Hz) oscillations. More surprisingly, strong eta (4 Hz) oscillations were also found in the majority of neurons, a phenomenon previously reported in the hippocampus [37]. These eta modulations are likely related to the phenomenon of theta skipping with respect to LFP observed in medial septum and MEC [46–48]. These findings were consistent with our model, and individual grid cells did indeed show oscillations in the range in which temporal jitter has the largest effect on toroidal topology. Furthermore, we found that the ratio of the power at the eta frequency to that of the theta frequency correlated with the critical jitter time that leads to the destruction of toroidal topology.

We also found that the eta-to-theta power ratio increased with increasing grid spacing. However, this increase in grid spacing, on its own, could not be the reason for the increased stability of the tori relative to modules with larger grid spacing. This is because first, Kang et al [14] in populations of Poisson spiking neurons show that decreasing grid spacing, while keeping everything else equal increases the chance of observing toroidal topology, defined as the presence of two long bars in $H_1$ (the author did not study $H_2$). In the experimental data, instead, it is harder to detect the torus in modules with smaller spacing. Furthermore, when we simulated populations of Poisson spiking cells with field size and spacing similar to those small modules, we consistently obtained a large degree of toroidality when we added oscillations with eta-to-theta ratio larger than what is seen in the data.

## Discussion

The critical role that oscillations play in the emergence of a robust toroidal topology can be attributed to a number of factors. Firstly, modulating rates by oscillations cause firing rates to be more correlated, decreasing the dimensionality of the phase space and thus making it easier for the toroidal topology to be evident. In fact, an initial dimensionality reduction through PCA before creating the barcodes is performed in the TDA approach used here and in [12] and oscillations may similarly reduce the effective dimensionality of the data before it is even further reduced by the PCA. More insight into the effect of oscillations was gained by applying persistent homology to different subsets of the data. Sub-sampling can be justified by computational constraints, but how to choose these samples may change the outcome [30–32]. While in real data, the toroidal topology was reported for samples with largest population activity [12], we found that this is not true for random samples. This was also the case for Poisson simulations with oscillations, but the opposite pattern was present without oscillations. And this difference persists even when firing rate in the simulations is increased. This results can be understood by noting that in the Poisson-spiking network without oscillations, high population activities occur because individual neurons randomly and independently emit more spikes than their mean; this variability destroys any pattern that could otherwise arise from the overlapping fields. Poisson simulations with oscillations differ from this: high activity time points are more likely to occur at peaks of the oscillations, where the spiking variability is more correlated and population vectors are more regular. This regularity together with regularities arising from the overlapping fields – which do not necessarily have to be idealized hexagonal patterns – seems to constitute an important cause of the toroidal topology in simulations with oscillations and in real data.

Oscillations are also known to decrease the variability of spiking of individual neurons in real data. For example they have been shown to reconcile rate-based and temporal coding in the case of phase precession in the hippocampus and entorhinal cortex [43]. Neuronal spiking in grid cells is affected by oscillations via the phenomenon of phase precession, and the way they lead to ordering of spikes emitted from different neurons [42]. Such ordering of spikes

can indeed also play a crucial role in the emergence of toroidal topology. However, although phase precession can be well quantified in one-dimensional tracks, this is much harder in open field 2D environments: it relies on a number of choices, e.g. the selection of high speed short segments of the trajectories through the fields [49–51]. A careful analysis of this issue is beyond the scope of this paper and we leave it to future study. However, we note that a good starting point might be to extend the sub-sampling comparison mentioned above, involving subsets of the population vectors in which spikes are ordered in a certain way.

One possible explanation of the appearance of toroidal topology is through path-integration in a continuous attractor network. Such networks generate the hexagonal firing pattern of grid cells and application of persistent homology to simulations of these networks yields toroidal topology [12]. However, these path-integrating models do not include the oscillatory dynamics that we have shown to be of crucial importance in the emergence of toroidal topology in experimental data. They also rely on specifically prescribed rate based dynamics formed by idealized connectivity patterns and show little tolerance to changes to such prescriptions [52–54]. It thus appears difficult to add oscillatory components to these models without adversely affecting their dynamics. The hexagonal patterns of grid cells could also primarily arise through feed-forward inputs [51,55–58]. Spiking neurons models implementing such feed-forward mechanism have been proposed [57] and feed-forward models that rely on oscillations from theta-modulated inputs have also been shown to generate the hexagonal firing pattern of grid cells [51,58]. Since our results show that oscillations and spatially periodic spiking patterns are sufficient to produce toroidal topology – without requiring recurrent connectivity – they better align with such feed-forward models compared to path-integration in continuous attractors. It is, however, important to note that assigning a primary role to feed-forward mechanisms in the formation of hexagonal patterns does not exclude an important role for the continuous attractor dynamics. Recurrent connectivity in MEC may still implement continuous attractors without path integration [59–63], adding stability to the hexagonal pattern which is primarily formed by other mechanisms [56,64].

While recordings from Wagon Wheel (WW) environments and during sleep, also exhibit some evidence of toroidal topology [12], in this paper, we only focused on recordings from the open field (OF) for which more data were available.

The conclusions and scope of this article are thus limited to mechanisms in OF and are not claimed to be universal. A quantitative comparison of the degree of toroidality between these different recording conditions may, however, better disentangle the comparative role of different factors that can, in principle, contribute to toroidal topology. These mechanisms include the hexagonal organization of the fields and oscillations. They also include factors such as spatial path-integration and the recurrent connectivity mentioned before, as well as animal's trajectory and the shape of the environment. In particular, regarding the latter, we note that the WW environment itself contains circles; traversing this environment may thus induce circular features on the population vector which will influence and be reflected in the barcodes. An in-depth analysis of this issue is beyond the scope of this paper.

Given the presence of oscillations in most areas of the brain, the approach used here and the role of oscillatory mechanisms in forming the topological features that we reported are likely to generalize beyond grid cells and toroidal topology. In general, TDA is a promising new tool for understanding the properties of high-dimensional data and the mechanisms that underlie such properties. However, the various choices and parameters involved in applying TDA to real data and the statistical fluctuations in such data make it necessary to quantitatively test the link between neuronal mechanisms and low-dimensional topological structures of the population activity. In this paper, this was done by introducing the quantity $\Gamma$ and studying its relationship to various features in the data from real or simulated grid cells.

A better theoretical understanding of how the statistical properties of $\Gamma$ relate to the structure of correlations in high-dimensional data could be a fruitful next step. This can be done, e.g., for data from distributions with known properties, and will contribute to an expanding effort for understanding statistical properties of persistent homological features [33,34,65].

## Methods

### Experimental procedures and preprocessing

Methods related to rats' breeding, electrode implantation, surgery and experimental procedures such as recordings and behaviors have been described in Gardner et al. [12], which provided the spike time data from cells classified as grid cells. The following is a brief summary of the main experimental procedures. Data were collected from three Long Evans rats (rats Q, R and S) using Neuropixels targeting the MEC-parasubiculum (PaS) region in four recording sessions. Data from the open field (OF) random foraging task in a 1.5×1.5 m arena when the rat had a speed exceeding 2.5 cm/s were used for topological analysis. We denoted each module with the name of the rat subscripted by the average grid spacing over the neurons in that module. Table 1 shows the correspondence between the names we used here and in Gardner et al. 2022 [12]. For the purpose of TDA, these data were preprocessed in the same way as Gardner et al. 2022 [12] and each available cell classified as a grid cell has been included in the analysis, unless stated otherwise. In particular, spike train data were preprocessed by converting spike times into delta functions, which were then smoothed using a Gaussian kernel. The smoothed activity was then binned at 10 ms and the population vector at every 5-th time bin was taken. From these population vectors, the 15000 most active population vectors were used for subsequent analyses. We analyzed the effect of this choice, comparing it with the same number of population vectors selected randomly in S1 Appendix.

Subsequently, principal component analysis (PCA) was applied to the selected population vectors, reducing the dimensionality of the data to six. A further downsampling technique was later used to reduce the point cloud to 1,200 points based on point-cloud density and neighborhood strength. The topological analysis, yielding the barcodes described in the next subsection, was then applied to analyze this reduced point cloud.

### Topological data analysis (TDA) and barcodes

All topological analyses were performed separately for each module, in each recording session. Persistent homology was used to find the low-dimensional representation of neural activity. Briefly, the algorithm starts by considering spheres of small, equal radii around each data point in a high-dimensional space. In the beginning, the radius is so small that these spheres do not overlap. As the radius values are increased, the spheres start to overlap. If, over a range of radii, the overlapping spheres form a d-dimensional hole, a bar is added to the barcode in dimension d, indicating the start and end points of the corresponding range. Thus, the holes corresponding to longer bars are more persistent and likely to represent topological features of the data in high dimensions. For toroidal topology, the relevant barcodes are those in dimensions 1, 2 and 3, indicated by $H_0$, $H_1$ and $H_2$. Since the $H_0$ barcode shows only one long bar in every analyzed set, we only show $H_1$ and $H_2$ in the figures. The software package Ripser was used for all computations of persistent cohomology. For the toroidal visualization the non-linear dimensionality reduction algorithm UMAP was used with the following parameters: 'n_neighbors' =1000, 'min_dist' =0.5, 'n_components'=3, metric='cosine'.

**Computation of the degree of toroidality.** The d-th component of the degree of toroidality is defined in Eq. (1) and is reproduced below

$$\Gamma_d^{ref} = 1 - \widehat{d}_B\left(\tau_d, \tau_d^{ref}\right) \tag{5}$$

where $\widehat{d}_B$ is the normalized bottleneck distance and $\tau_d^{ref}$ is the barcode of the reference torus in dimension d. In the following, we describe the computation and characteristics of this measure.

The bottleneck distance $d_B(P, Q)$ is a common distance measure to compare two sets of bars $P$ and $Q$. Suppose that $p$ is a bar in the barcode $P$ starting at $x_p$ and ending in $x'_p$, and similarly that $q$ is a barcode in $Q$ starting at $x_q$ and ending at $x'_q$. Defining

$$\|p - q\|_\infty = \max\left(|x_p - x'_p|, |x_q - x'_q|\right) \tag{6}$$

we have

$$d_B(P, Q) = \inf_f \sup_{p \in P} \|p - f(p)\|_\infty \tag{7}$$

where $f: P \to Q$ is a bijective. Intuitively, for each matching assignment $f$, that is for each bar $p$ matched to a bar $f(p)$, one first calculates the distance as the largest of the differences between the starting points and end points of $p$ and $f(p)$. The bottleneck distance is the smallest value of this quantity amongst all mappings $f$. In other words, it corresponds to the matching of the bars for which the maximum distance of the end point and start points of corresponding bars in $P$ and $Q$ is minimal. The first problem with the bottleneck distance, as is clear from Eq. (6) and (7) is that its value depends on the scale of the barcodes, which in turn reflects how far points are in high-dimensional space, according to the metric used in calculating the barcodes. As such, the maximum achievable value is not fixed. To avoid this problem in defining the degree of toroidality, we therefore build on this distance by applying it to normalized barcodes such that the maximum distance in each barcode is unitary. This is done by dividing the barcodes by

$$u(P) = \max_{p, p' \in P} \|p - p'\|_\infty \tag{8}$$

which ensures that $d_B(P/u(P), Q/u(Q))$ is between zero and one. As already shown in Fig 1, this normalization resolves the problem of the relative scales of bars in a barcode.

## Parametrization of the 6-dimensional torus

The torus in 6 dimensions reported in Fig 2 is described by

$$\begin{aligned}
p_1 &= C_1 \cos(a_1 u + b_1 v) \\
p_2 &= C_1 \sin(a_1 u + b_1 v) \\
p_3 &= C_2 \cos(a_2 u + b_2 v) \\
p_4 &= C_2 \sin(a_2 u + b_2 v) \\
p_5 &= C_3 \cos(b_3 v) \\
p_6 &= C_3 \sin(b_3 v)
\end{aligned} \tag{9}$$

where $u, v \in [0, 2\pi)$ are the angular coordinates. The parameters are set to the following values: $a_i = 1$, $b_1 = 1/\sqrt{3}$, $b_2 = -b_1$, $b_3 = 1$ and $C_i = 1$.

## Estimating time scales

The estimation of the critical time scale $\Delta t_C^{(1)}$ was performed by fitting with a least squares method a sigmoid function $s(\Delta t)$ to the value of $\Gamma_1$ as a function of the temporal jitter magnitude $\Delta t$:

$$s(\Delta t) = \frac{L}{1 + \exp\left(-k(\Delta t - \Delta t_C^{(1)})\right)} + b \tag{10}$$

We detected the inflection point of the sigmoid $\Delta t_C^{(1)}$. We repeated the same process separately for $\Gamma_2$, detecting the value $\Delta t_C^{(2)}$. We designated $\Delta t_C$ as the minimal value of these two inflection points, $\Delta t_C = \min\left(\Delta t_C^{(1)}, \Delta t_C^{(2)}\right)$. This is a lower bound on the temporal jitter that destroys toroidality, and the values are reported in Table 2.

We estimated the *average behavioral time scale* as the ratio between grid spacing and average speed of the rat for each module. This ranges from 13.1 cm/s in session 'day 2' of rat R to 16.1 cm/s in session 'day 1' of the same rat and thus yields the *average behavioral time scale* to range from 3.8 s for module $R_{61}^1$ to 8.0 s for module $R_{105}^3$. This is indeed several orders of magnitude larger than the critical timescale for each module. It is to be noted that the *average behavioral time scale* is an underestimation of the actual average time that the rat takes to go from one field to the other, because it assumes that the rat is running on a straight line.

## Parameters of the oscillations

Unless otherwise stated in the text, for the simulations reported in Results, we have set $\lambda_0 = 0.05$, $G_0 = 1.5$, $x_0 = 0.4$ in Eqs. (3) and (4), the network consisted of $N = 75$ neurons and the simulations were run over half the trajectory of the first recording (day 1) of rat R. We also set $c_1 = 1$, $c_2 = 0$ for the case of simulations without oscillations and $c_1 = 1$, $c_2 \neq 0$ when oscillations were added. Specifically, this was done by considering $m = 200$ oscillators in Eq. (3) with $\omega_\mu$ spaced logarithmically in the interval $[1, 50]$ Hz and $A(\omega_\mu) = 0.25\omega_\mu^{-1/2}$. The oscillators with $\omega_\mu = 4, 8$ Hz have $A(\omega_\mu) = 0.5\omega_\mu^{-1/2}, 0.8\omega_\mu^{-1/2}$, respectively. We also chose $c_1 = 0$ and $c_2 = 0.5884$; the latter choice was made such that the number of spikes emitted by each neuron is similar to that of the mean of the neurons in $R_{85}^2$ and that the removal of the oscillations ($c_1 = 1$ and $c_2 = 0$) does not change this.

The value of $c_2$ in the case with oscillations was chosen so that, with all other parameters equal, the average number of spikes would not change with respect to the simulations without oscillations. This was done by assuming that at each spatial position, each phase of each oscillator is observed many times, that is

$$c_2^{-1} = \lim_{T \to \infty} \frac{1}{T} \int_0^T dt \left[\sum_{i=0}^m A(\omega_\mu) \sin(2\pi\omega_\mu t)\right]_+ \tag{11}$$

In practice, this was computed for any choice of the oscillatory amplitude by assuming $T = 36$ and discretising the integral in steps of 0.001. The results was not affected by small changes to these choices. As expected, setting $c_2$ in this way, the average number of spikes emitted did not vary between the cases with oscillation and without.

## Computation of power spectral density (PSD)

PSD of each individual grid cell was calculated separately using FFT (numpy function fft.fft()). To allow comparison between different datasets, the FFT computation for all data was restricted to the recording length with the shortest duration. The PSD thus obtained over the same duration of the data, were normalized by the total power in the frequency range [0.1, 500] Hz to compensate for the differences in firing rates of neurons. These PSDs were then averaged over all the simultaneously recorded units in each module. Similar analyzes were performed when the rat was running at speed faster than 2.5 cm/s. Qualitatively similar results were obtained using both of these analysis methods. The eta power was calculated in the following range for each module based on the trough in the corresponding PSDs: [2.5, 5.9] Hz for modules $R_{85}^2$ and $R_{121}^3$, [2.5, 5.2] Hz for $R_{58}^1$ and $R_{79}^2$, [2.9, 6] Hz for $Q_{70}^1$ and [2.7, 6.5] Hz for $Q_{99}^2$. The theta power was calculated in the following range for each module: [5.9, 11] Hz for modules $R_{85}^2$ and $R_{121}^3$, [5.2, 11] Hz and [5.2, 10.5] for $R_{58}^1$ and $R_{79}^2$, [6, 12.9] Hz for $Q_{70}^1$ and [6.5, 12.9] Hz for $Q_{99}^2$.

## Supporting information

**S1 Appendix. Persistent homology and the choice of time points and population vectors.** (PDF)

**S1 Fig. $\Gamma$ as a function of the number of cells.** Each plot shows the increase of $\Gamma_1$ and $\Gamma_2$ with the number of cells, when all recorded cells are included in the analysis. (TIF)

**S2 Fig. $\Gamma$ as a function of the number of pure grid cells.** Each plot shows the increase of $\Gamma_1$ and $\Gamma_2$ with the number of cells, when pure grid cells only are included in the analysis. (TIF)

**S3 Fig. Toroidality has a sigmoidal dependence on the magnitude of temporal jitter over a range in which hexagonality is maintained.** Everything is the same as in Fig 6, but for modules $R_{85}^2$, $R_{79}^2$ and $Q_{70}^1$. (TIF)

**S4 Fig. Dependence of $\Gamma$ on oscillation frequency with smaller spacing.** The same simulation as the one in Fig 10A and 10B with smaller spacing (similar to $R_{58}^1$) is shown. (TIF)

**S5 Fig. Barcodes consistent with toroidal topology from a model with neurons having spatial fields on square lattice.** The same simulation as the one in Fig 15 when eta and theta oscillations are introduced for a square grid cell module. (TIF)

**S6 Fig. Dependence of $\Gamma$ on firing rate and $\sigma$ for N=150.** The same simulation as the one in Fig 8 with $G_0 = 0.8$, $G_0 = 1.5$ and $G_0 = 3$ is shown for $N = 150$. (TIF)

**S7 Fig. Dependence of spectral power bands on critical jitter.** Same as Fig 14 but for $A_\eta$ and $A_\theta$ independently, as well as for $A_\delta/A_\eta$ and $A_\delta/A_\theta$. (TIF)

## Acknowledgments

The authors are most grateful to Mayank Mehta for helping with the conceptualization, and interpretation of the results at an earlier stage of this work. G.d.S. and Y.R. are also thankful for fruitful discussions with Benjamin Dunn and Mauro M. Monsalve-Mercado.

## Author contributions

**Conceptualization:** Giovanni di Sarra, Siddharth Jha, Yasser Roudi.

**Formal analysis:** Giovanni di Sarra, Yasser Roudi.

**Investigation:** Giovanni di Sarra.

**Methodology:** Giovanni di Sarra.

**Software:** Giovanni di Sarra, Yasser Roudi.

**Supervision:** Yasser Roudi.

**Validation:** Giovanni di Sarra.

**Visualization:** Giovanni di Sarra.

**Writing – original draft:** Giovanni di Sarra, Siddharth Jha, Yasser Roudi.

**Writing – review & editing:** Giovanni di Sarra, Yasser Roudi.

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
