## [Decision Letter · Decision Letter 0]

27 Jul 2024

Dear Dr. Roudi,

Thank you very much for submitting your manuscript "The role of oscillations in grid cells’ toroidal topology" for consideration at PLOS Computational Biology.

As with all papers reviewed by the journal, your manuscript was reviewed by members of the editorial board and by several independent reviewers. In light of the reviews (below this email), we would like to invite the resubmission of a significantly-revised version that takes into account the reviewers' comments.

In particular, it will be essential to clarify whether the proposed oscillation mechanism is mostly useful to compensate for the stochastic spiking of the Poisson simulations (reviewer 1), and how it would compare to CAN models (reviewer 3).

We cannot make any decision about publication until we have seen the revised manuscript and your response to the reviewers' comments. Your revised manuscript is also likely to be sent to reviewers for further evaluation.

Sincerely,

Timothée Proix

Guest Editor

PLOS Computational Biology

Andrea E. Martin

Section Editor

PLOS Computational Biology

Reviewer's Responses to Questions

**Comments to the Authors:**

Reviewer #1: When considered in neural state space, population responses from a medial entorhinal grid cell module lie on the surface of a torus. This toroidal structure has emerged as an important signature of grid cells. Such toroidal structure naturally emerges from models, but most modeling focuses on rate-based architectures and does not consider the role of spike timing in producing the population structure. In this study, the authors combine experimental data analysis and modeling to show that the synchrony of spikes in the 100-500ms time range is necessary for producing the toroidal topology seen in data.

I thought this a conceptually well-organized, well-written and thorough study that convincingly makes the case for thinking about spike timing and oscillations in the grid cell system and that points to the need for richer models. I appreciated the combination of data analysis and modeling and also thought the use of barcode distance was a promising way of probing structure that may find wider use (despite some concerns about the noise sensitivity of barcodes). I did not run the code, but based on a quick glance it seems well-organized and commented (nice job on that!), which greatly increases the chance that future work will build on these analyses.

My comments are primarily aimed at clarifying the description and interpretation of what the authors did.

1) In terms of interpretation, the authors (very nicely) highlight that rate maps of regular fields should give rise to toroidal topology but that spiking neurons with irregular fields may not. I particularly appreciated the Poisson simulations, which to me imply that stochastic spiking would be the main cause of the lack of toroidality, and thus that mechanisms to reduce variability in spiking are important to produce toroidality. I had some questions/comments related to clarifying this interpretation:

a) If stochastic spiking is the main issue, would the importance of oscillations and sensitivity to jitter decrease if neurons were to fire at higher rates? A simple simulation showing this in the model would be useful.

b) Do the Poisson simulations (which have regular firing fields) imply that irregular fields are not an important cause of the lack of toroidality? It would be beneficial to the field if the authors could rule out some mechanisms in addition to pointing to the importance of others.

c) Is there an effect of the number of neurons? For example, do the authors predict that modules with weak toroidality would show strong toroidality if more neurons were recorded?

2) I was confused as to the reference torus in Fig. 2 and whether it was a good probe for toroidality. As described, the reference torus here involves keeping the two longest bars in H1 and the longest in H2. I see why this might be a good reference to compare jittered data to original data (under the assumption that the original data is toroidal). But it may not be a good way to evaluate the toroidality of the data itself. E.g., say I had data from a plane, calculated its barcodes, kept its two longest bars in H1 and longest bar in H2 (which would all be quite short), and then treated this structure as the reference. In that case I might conclude that the dataset was “toroidal” based on similarity to my reference barcodes. Shouldn’t some measure of relative length/persistence of these bars factor into the degree of toroidality of these populations? Relatedly, if the reference is constructed from the experimental data itself, then why is the toroidality of the experimental data not higher? Is this entirely due to differences in shorter bars (that are set to minimum length in reference)? I think I must be missing something about the reference here.

3) It would be useful to support the observation that the jitter timescales are “several times smaller than the time it takes for the rat to go from one grid field to another, which is of the order of seconds” given that smearing spikes across firing fields would obviously degrade toroidality. More specifically, given the rats’ velocity, what fraction of a firing field width have they moved over the critical timescale of jitters. Does moving locations on the jitter timescale contribute at all to the lack of toroidality?

4) Sentence on Pg. 8 “Both these indicate that the emerging toroidal topology is much more sensitive to variability in spike times than the smaller ones.” The previous lines made me think that the smaller modules were more sensitive but this sentence seems to contradict that. If not a typo it would be good to expand on the point/clarify.

5) Fig. 4: Is there a reason why critical jitter is calculated using Γ1 for first two modules and Γ2 for the third?

6) Eq. 3: Uses the same index for neurons and for temporal frequencies (i). Shouldn’t these be different? Could also briefly unpack the sum of oscillation frequencies term in the equation for the reader. In the methods could also describe the normalization of A(omega) and the range and spacing of omegas chosen.

7) What are the values of firing rates for simulations in the center of a firing field and how do they compare to data? Relatedly, it would be useful to add a scale to the firing rate maps throughout, including in the simulations (especially if the effect is dependent on firing rate).

8) Fig. 6: what is the value of sigma that best fits the experimental data? Is the best match between simulations and data at this value of sigma, or does it continue to increase with bigger sigma? If it does continue to increase, why is this?

9) I found the results on eta-to-theta ratio somewhat confusing. The authors show in simulations and figures that including oscillations in these frequency bands improves toroidality. However, these results seem disconnected from results on the ratio of eta to theta power. As far as I can tell, Fig. 11 is showing that including these oscillations improves toroidality but not showing that these oscillations need to have a high ratio or modulating the ratio. It was also not clear why mechanistically the ratio of these oscillations should be the important feature for toroidality (rather than simply oscillations in these power bounds). The Discussion seems to hint that these results are being driven by changes in the eta power (which would then change the eta to theta ratio, assuming that theta did not also vary similarly in a way that kept the ratio the same). If so that could be made more explicit and might be a simpler framing.

10) For completeness, some more information on how the TDA was performed could be included in methods (I realize that the authors follow Gardner et al.). For example, the PCA preprocessing step is briefly mentioned in main text but not in Methods.

Reviewer #2: In thsi manuscript, di Sarra et al. present a novel framework to understand the relationship between grid cell spatial patterns of activity and the description of such activity as living on a toroidal manifold. The main claim of the paper is that these two properties associated with grid cells are mostly independent, being based on the organization of activity over largely separated time-scales (seconds vs. 100-400ms). In particular the authors claim that the emergence of a toroidal topology explaining the relative configuration of population activity patterns can be ascribed to the modulatory effects on average activity of oscillations in the thera and eta band of the spectrum. When spiking activity is subject to the combination of thse two oscillations, toroidaility is found to be strengthened in a model of grid cell activity, while a strong modulation in these two bands in experimental data is associated with stronger evidence of a toroidal topology. Importantly, hexagonal arrangment of fields seem to be mostly a separated property.

In my opinion the authors present two main contributions. The first one is to define a measure of proximity to a toroidal topology. The toroidal hypothesis is gathering momentum in the study of grid cells, and with it the reliance on topological methods to define the properties of this neural population. Most of this evidence, though, is based on a significance-based assessment, that is a zero or one measure of compatibility between the neural activity and a toroidal arrangment of the activity patterns. Such dichotomous approach risks to mask the variability in the arrangement of activity and to lead to an oversimplification or a distortion of the actual richness in activity structures. Therefire I find the introduction of the similarity measure quite important as it allows to capture the proximity and the degree of proximity to a torus, providing information on the appropriateness of the description. Such measure also allows to more precisely measure the contribution of specific features of neural activity in driving the system towards or away from toroidailtiy.

Accordingly, the second contribution of the paper, is to explicitely investigate which aspects of neural activity can be identified as most relevant for approaching a toroidal topology. The separation between 'spatial coding' (in this case the hexagonality of the field arrangment) and 'temporal coding' (the modulation of activity at timescales generally shorter than the behavioral one, in most of the cases involving theta oscillations) is quite popular when it comes to the study of the hippocampus and neighboring cortices, but it has never been applied to this problem.

The results of the paper are in many senses surprising, as they seem to dissociate the presence of hexagonal firing and that of torodaility. The analysis seems to be still mostly at the qualitative level, as there is no precise mechanism proposed in the paper to explain the role of oscillations. At the same time both the experimental and the model analysis are quite convincing in making the case for a prominent role of oscillations in the emergence of toroidaility.

To fully appreciate the relevance of the approach and of the model though, I think that a number of additional analyses would be helpful.

1. My first point regards the role of oscillations: increased coherence in spikes is mentioned as their role in shaping the torus, but why 2 oscillations? Does it work only with one modulated band? And is there anythig specific about these two bands? It would be very useful to further contrast the behavior of the model by using only one of the two oscillations and by changing their frequency.

2. Hexagonal arrangement is shown to be not sufficient for having a torus. But is it necessary? It would be greatly informative to run simulations replacing grid cells with units having multiple fileds spanning the environment and mantaing a similar distance from each other, but lacking the precise arrangement of an hexagonal tesselation. Parallely it would also be interensting to test the degree of torodality in the non-grid population in the dataset. Figure 2 seems to suggest that the inclusion of non-grid cells is not affecting the torus, and if anything is increasing the degree of torodality. This would indicate that each of the populations has an high degree of torodality.

3. Putting these two points together, I also wonder wether higher degree in small grids could be achieved by using higher frequency ranges, allowing for a better separation between temporal activity and the behavioral scale of field crossing.

Reviewer #3: In the submission “The role of oscillations in grid cells’ toroidal topology” the authors re-analyze data from Gardener et al. 2020 and, by using simulations, investigate possible contributions of oscillations to the toroidal topology. In this manuscript, a metric for the degree of toroidality based on persistence homology is introduced, and temporal jitters, together with simulations, are used to assess the robustness of toroidal topology. The manuscript is timely, and it’s topic of potential interest and within the scope of the Journal. Although the motivation and writing of the paper are clear, the methods are not always described clearly enough to fully assess the validity of the simulations and, therefore, conclusions drawn from them.

The results presented in the manuscript suggest a potential contribution of oscillations to toroidal stability. However, while discussing CAN models versus weighted feedforward, input-driven networks, it is less clear how these results favor one or the other. The two points made on the bottom of page 16, could be made without any of the results described in the manuscript. The fact that an alternative model leads to similar toroidal topography demonstrates a possibility but does not rule out CAN models. As later addressed in the manuscript, attractor dynamics do play an essential role in grid cell populations. Therefore, the above-mentioned comparison of models needs either to be toned down or seamlessly related to the manuscript’s results. Please find my specific comments below.

MAJOR:

1.) Parameters of the simulations should be stated more clearly, both in the results section (page 4 ), and in the methods sections; at least i.) how many neurons were simulated, ii.) how many realizations with what parameters, iii.) whether the oscillations in eq. 3 were synchronous for all neurons simulated.

2.) Although it seems to be intended in the simulations, one could argue that the oscillatory part in eq. 3 limits the temporal variability of neuronal firing, increasing correlation within the simulated population and hence facilitating toroidal topology, given the presence of repetitive firing fields. The key question is whether any other mechanism, e.g., organized, sequential firing, that limits the temporal variability between neurons would not have the same effect.

3.) The sampling of binned spike counts needs to be stated explicitly; Gardner et al. 2022 used 250ms intervals between samples, which would be in the range of the critical jitter times described in the manuscript. It would be relevant to rule out whether this binning had any influence on the results reported.

4.) On page 3, there is mention that the absolute power of theta or eta power did not correlate with critical time scales, but these results are not included in the manuscript, nor is whether the relative power of eta or theta (to population average) might play a key role.

5.) Given the proposed relevance of oscillations, does the amount of i.) theta-modulated cells or ii) cells showing phase-precession impact toroidal topology?

6.) The numbers in Table 2 might be incomplete; from my understanding, module spacing and critical jitter times should be listed here.

7.) The manuscript sometimes reads like a commentary on Gardner et al. 2022, especially on page 6 when discussing the results of Extended Fig. 6A or when commenting on the Wagon Wheel environments on page 16. Usually, present results are discussed in the context of previous literature. I would suggest limiting these comments or discussions to the relevance of the present results of the submission.

MINOR:

A.) The second to last sentence of the abstract is hard to understand. Consider re-writing or splitting it into two sentences.

B.) The introduction presents not only the context and motivation of the study but also partly discusses the results, which I found unusual for an introduction. I would consider limiting to context, motivation, and a brief outline of the study results.

C.) Given the similarity in Fig.1 a and e, why is the x-axis in e) not matching the one in a) ?

D.) For a better illustration of the comparisons following Fig. 2, would a difference plot (all minus pure grid cells) help to show the relative in-/decrease of tau parameters in 2D?

E.) It would be helpful to add the integration time constant and time between grid fields on page 8, such as in the discussion (pg.15).

F.) From my understanding, the peaks shown in Fig. 5 at 4 and 8 Hz in the PSD only show that the simulation worked. As of now, it reads like a result; it would be surprising if we did not see peaks here, given the oscillations included in the simulations. Please clarify or mention these are peaks confirming the functionality of the simulations.

G.) Is the correlation mentioned in Fig. 10b computed from N=6 values? Please state explicitly in the caption.

**Have the authors made all data and (if applicable) computational code underlying the findings in their manuscript fully available?**

Reviewer #1: Yes

Reviewer #2: Yes

Reviewer #3: Yes

PLOS authors have the option to publish the peer review history of their article (what does this mean?). If published, this will include your full peer review and any attached files.

Reviewer #1: No

Reviewer #2: **Yes: **Federico Stella

Reviewer #3: No
---

## [Decision Letter · Decision Letter 1]

7 Jan 2025

Dear Dr. Roudi,

We are pleased to inform you that your manuscript 'The role of oscillations in grid cells’ toroidal topology' has been provisionally accepted for publication in PLOS Computational Biology.

Best regards,

Timothée Proix

Academic Editor

PLOS Computational Biology

Andrea E. Martin

Section Editor

PLOS Computational Biology

Reviewer's Responses to Questions

**Comments to the Authors:**

Reviewer #1: I thank the authors for their thorough responses to my comments, which have resolved my concerns.

Reviewer #2: The authors dealt with all the points I have risen and I appreciate their thorough responses.

I consider the manuscript suitable for publication.

Reviewer #3: All my comments were addressed appropriately.

**Have the authors made all data and (if applicable) computational code underlying the findings in their manuscript fully available?**

Reviewer #1: Yes

Reviewer #2: None

Reviewer #3: Yes

PLOS authors have the option to publish the peer review history of their article (what does this mean?). If published, this will include your full peer review and any attached files.

Reviewer #1: No

Reviewer #2: **Yes: **Federico Stella

Reviewer #3: No

---

## [Editor Report · Acceptance letter]

PCOMPBIOL-D-24-00753R1

The role of oscillations in grid cells’ toroidal topology

Dear Dr Roudi,

I am pleased to inform you that your manuscript has been formally accepted for publication in PLOS Computational Biology. Your manuscript is now with our production department and you will be notified of the publication date in due course.

With kind regards,

Olena Szabo
